# Structural basis for antibacterial peptide self-immunity by the bacterial ABC transporter McjD

Kiran Bountra[1,2,†], Gregor Hagelueken[3,†], Hassanul G Choudhury[1,2,†], Valentina Corradi[4], Kamel El Omari[2,5], Armin Wagner[2,5], Indran Mathavan[1,2], Séverine Zirah[6], Weixiao Yuan Wahlgren[1,2,7], D Peter Tieleman[4], Olav Schiemann[3], Sylvie Rebuffat[6] & Konstantinos Beis[1,2,*] ID

## Abstract

Certain pathogenic bacteria produce and release toxic peptides to ensure either nutrient availability or evasion from the immune system. These peptides are also toxic to the producing bacteria that utilize dedicated ABC transporters to provide self-immunity. The ABC transporter McjD exports the antibacterial peptide MccJ25 in *Escherichia coli*. Our previously determined McjD structure provided some mechanistic insights into antibacterial peptide efflux. In this study, we have determined its structure in a novel conformation, apo inward-occluded and a new nucleotide-bound state, high-energy outward-occluded intermediate state, with a defined ligand binding cavity. Predictive cysteine cross-linking in *E. coli* membranes and PELDOR measurements along the transport cycle indicate that McjD does not undergo major conformational changes as previously proposed for multi-drug ABC exporters. Combined with transport assays and molecular dynamics simulations, we propose a novel mechanism for toxic peptide ABC exporters that only requires the transient opening of the cavity for release of the peptide. We propose that shielding of the cavity ensures that the transporter is available to export the newly synthesized peptides, preventing toxic-level build-up.

**Keywords** antibacterial peptide ABC transporter; membrane protein; molecular dynamics; PELDOR; transporter structure

**Subject Categories** Membrane & Intracellular Transport; Microbiology, Virology & Host Pathogen Interaction; Structural Biology

The EMBO Journal (2017) 36: 3062–3079

## Introduction

ATP-binding cassette (ABC) transporters (exporters) are one of the largest superfamilies of membrane transporters and are found in both bacteria and eukaryotes (Holland *et al*, 2003); they are usually involved in efflux or detoxification pathways including multi-drug resistance in both bacterial and eukaryotic cells. Bacteria, under nutrient starvation, produce and release antibacterial peptides, microcins, which can kill microcin-sensitive cells and therefore provide more nutrients for the surviving bacteria (Duquesne *et al*, 2007a). The microcin J25 (MccJ25) is a plasmid-encoded, ribosomally synthesized and post-translationally modified antibacterial peptide consisting of 21 amino acids and adopting a lasso structure (Rosengren *et al*, 2003); its C-terminal tail threads through an N-terminal eight-residue macrolactam ring, where it is locked by bulky amino acid side chains, forming a compact interlocked structure. Four genes are required for the biosynthesis and export of MccJ25 (Duquesne *et al*, 2007b): *mcjA* encodes the linear precursor of MccJ25; *mcjB* and *mcjC* encode enzymes involved in the post-translational modification of McjA to the lasso structure; and *mcjD* encodes an ABC exporter that is required for both MccJ25 secretion and self-immunity of the producing strain towards MccJ25 (Choudhury *et al*, 2014). Since the genes are under the same operon, production and export from the cell are very efficient and toxic levels of MccJ25 are prevented from building up. Inside the target cell (Runti *et al*, 2013; Mathavan *et al*, 2014), MccJ25 inhibits the bacterial RNA polymerase (Semenova *et al*, 2005).

We have previously functionally characterized and determined the high-resolution structure of McjD from *Escherichia coli* (Choudhury *et al*, 2014; Fig 1A). Its core architecture is composed

1  Department of Life Sciences, Imperial College London, London, UK
2  Rutherford Appleton Laboratory, Research Complex at Harwell, Oxfordshire, UK
3  Institute for Physical and Theoretical Chemistry, University of Bonn, Bonn, Germany
4  Centre for Molecular Simulation and Department of Biological Sciences, University of Calgary, Calgary, AB, Canada
5  Diamond Light Source, Oxfordshire, UK
6  Communication Molecules and Adaptation of Microorganisms Laboratory (MCAM, UMR 7245 CNRS-MNHN), Muséum National d'Histoire Naturelle, Centre National de la Recherche Scientifique, Sorbonne Universités, Paris, France
7  Chemistry & Molecular Biology, University of Gothenburg, Göteborg, Sweden
   *Corresponding author. Tel: +44 1235 567809; E-mail: kbeis@imperial.ac.uk
   †These authors contributed equally to this work

of a dimeric transmembrane domain (TMD) of 12 transmembrane (TM) helices, which forms the translocation pathway across the membrane bilayer and contains the ligand binding site, and a dimeric nucleotide binding domain (NBD) where ATP binds and is hydrolysed. ABC exporters either use the alternating access mechanism or outward-only mechanism (Perez *et al*, 2015) to transport their substrates. ABC exporters that use the alternating access mechanism switch between inward- and outward-facing states, which exposes the ligand binding site alternatively to the inside or outside of the membrane, coupled to ATP binding and hydrolysis (Beis, 2015). The previous structure of McjD was determined in a nucleotide-bound outward-occluded conformation (occluded at both sides of the membrane), representing an intermediate state between the outward- and the inward-facing conformations (Choudhury *et al*, 2014).

In order to understand the detailed mechanism of toxic peptide export by bacterial cells, it is important to trap the transporter in different conformations. Here, we have determined the structure of McjD in a novel conformation, apo inward-occluded and an additional nucleotide-bound state, high-energy outward-occluded intermediate with bound ATP-vanadate ($ADP-VO_4$). We further characterized these new states in *E. coli* membranes using predictive cysteine cross-linking in inside-out vesicles (ISOVs). Using a spin-labelled McjD mutant and pulsed electron-electron double resonance (PELDOR), also known as DEER (double electron-electron resonance), the conformation of McjD was investigated in bicelles. In addition, we applied molecular dynamics (MD) simulations to study the dynamics of the outward-occluded conformation of McjD in the presence of ATP or ADP molecules bound at the NDBs. The new conformations in combination with the PELDOR data, transport assays in proteoliposomes and molecular dynamics simulations have significant mechanistic implications in understanding the detailed mechanism of the antibacterial peptide exporter McjD.

# Results

## Structures of McjD in different conformations

In order to gain a detailed understanding of the conformations that McjD adopts during the transport cycle, we have determined its structure in the presence of the transition state analogue $ADP-VO_4$ (mimic of ATP hydrolysis) and in the absence of nucleotides, that is in the apo form, at 3.4 Å and 4.7 Å resolution, respectively (Fig 1B and C). We determined the structure of McjD in complex with $ADP-VO_4$ by molecular replacement using the McjD-AMPPNP (adenosine 5′-(β,γ-imido)triphosphate) (PDB ID: 4PL0) structure as a search model (Choudhury *et al*, 2014). The structure was refined with an $R_{work}$ of 25.7% and $R_{free}$ of 26.0%. Clear electron density was observed for $ADP-VO_4$ and $Mg^{2+}$ (Fig EV1A–D). The presence of vanadate was verified by collecting the data close to the vanadium K-edge, 2.26 Å wavelength and calculating anomalous difference electron density maps (see Materials and Methods; Fig EV1C). At both NBDs, two strong electron density peaks of 14 and 20 sigma, respectively, confirmed the presence of two $VO_4$ groups. McjD-ADP-$VO_4$ is almost identical to the McjD-AMPPNP, as the two structures can be superimposed with a root-mean-square deviation (rmsd) of

0.7 Å over 560 Cα atoms (Fig EV2A and B). In contrast to the MsbA-ADP-$VO_4$ structure that adopts an outward-open conformation as a result of domain intertwining (Ward *et al*, 2007), the McjD-ADP-$VO_4$ structure is outward-occluded without domain intertwining (Fig 2A). We call it here high-energy intermediate outward-occluded. The TMD dimer interface in McjD-ADP-$VO_4$ is formed between TM2 and TM5/TM6 from one subunit with the equivalent TMs from the opposite subunit similar to the McjD-AMPPNP structure. In the presence of ADP-$VO_4$, the NBDs are dimerized with the P-loop and ABC signature motifs involved in the binding of the nucleotide. The McjD-ADP-$VO_4$ structure represents the transition state of a water molecule making a nucleophilic attack on the γ-phosphate of ATP.

The McjD-apo structure was also determined by molecular replacement using the McjD-AMPPNP structure but only after splitting the McjD monomer into two domains, TMD and NBD (see Materials and Methods). The structure was refined to an $R_{work}$ of 31.3% and $R_{free}$ of 33.4%. Clear electron density could be observed for both the TMD and NBDs (Fig 3A and B). Inspection of the NBDs did not reveal any electron density for nucleotides that may have been co-purified with McjD (Fig 3A). In the absence of nucleotides, ABC transporters adopt an inward conformation with their TMDs separated and NBDs disengaged (Ward *et al*, 2007; Fig 2B). A striking difference between McjD-apo and other apo ABC exporters, such as MsbA (Ward *et al*, 2007) and PglK (Perez *et al*, 2015), is that its TMD is in an occluded conformation, similar to the McjD-AMPPNP structure, whereas the NBDs have disengaged in the absence of nucleotides. This conformation is called here apo inward-occluded. Inward-facing ABC exporters display domain intertwining of TMs 1–3 and 6 from one subunit and TMs 4–5 from the opposite subunit, which results in opening of the TMD to the cytoplasmic side of the inner membrane (Ward *et al*, 2007; Perez *et al*, 2015). In the inward-occluded McjD, TMs 3–5 from one subunit have moved towards the equivalent TMs of the opposite subunit, resulting in the loss of intertwining and occlusion of the cytoplasmic opening (Figs 2B and EV2). The McjD-apo structure can be superimposed with the McjD-AMPPNP structure with an rmsd of 2.1 Å over 569 Cα atoms; their TMDs can be superimposed with an rmsd of 0.7 Å over 290 Cα atoms. The heterodimeric human sterol apo-ABCG5/8 (Lee *et al*, 2016) and *Pseudomonas aeruginosa* lipopolysaccharide apo-LptB₂FG (Luo *et al*, 2017) do not display intertwining either.

In the absence of nucleotides, the NBDs of ABC exporters disengage (Ward *et al*, 2007; Perez *et al*, 2015). The NBDs of the McjD-apo have separated by 7.9 Å (measured between S509 and S509') relative to the McjD-AMPPNP structure, a distance much shorter compared to MsbA-apo with a distance of 35 Å (Ward *et al*, 2007). The NBDs have moved in a "scissors-like" motion similar to the transition between inward and outward-open MsbA. A small degree of disengagement between the NBDs has also been reported for the heterodimeric ABC exporter TM287/288 (Hohl *et al*, 2012; Hohl *et al*, 2014), ABCG5/8 (Lee *et al*, 2016) and LptB₂FG (Luo *et al*, 2017).

## Cysteine cross-linking in ISOVs

The presence of apo inward-occluded and high-energy intermediate outward-occluded states have not been observed or characterized in

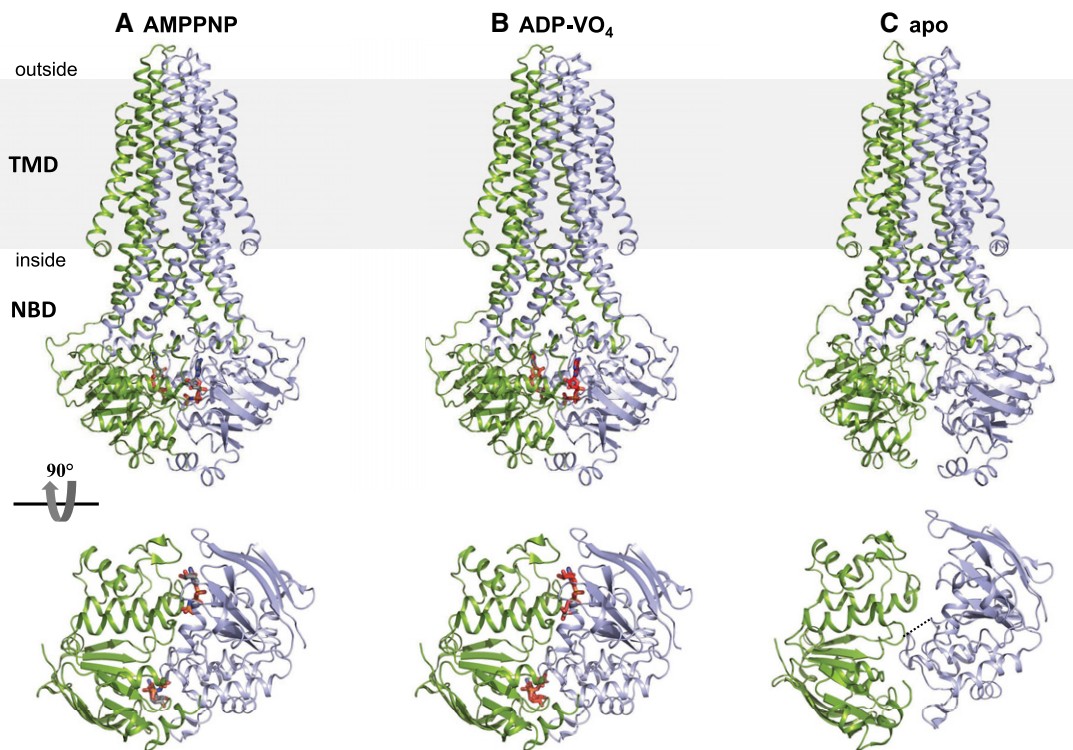

**Figure 1.  Crystal structures of McjD in three distinct conformations.**

A–C  McjD is shown in cartoon and nucleotides in red sticks. Each half transporter is coloured in blue and green. Top panel is a view along the plane of the membrane, and bottom panel shows the NBDs for each state. The membrane is depicted in grey. (A) AMPPNP-bound outward-occluded conformation (PDB ID: 4PL0; Choudhury *et al*, 2014), (B) high-energy intermediate outward-occluded conformation (ADP-VO$_4$) and (C) apo inward-occluded. In all three transport steps represented by the crystal structures, the TMDs are in an occluded conformation without access to the periplasmic (nucleotide-bound) or cytoplasmic side (apo) of the membrane. In the presence of nucleotides, the NBDs dimerize similar to other ABC exporters. In the apo state, the NBDs of McjD disengage but with a smaller degree of separation compared to other ABC exporters. The two NBDs have separated by 7.9 Å as indicated by a black dashed line.

other ABC exporters. We selected specific amino acids along the periplasmic, cytoplasmic and NBD sides to perform predictive cysteine cross-linking in ISOVs to further characterize them in their native lipid environment. We have previously shown that the occluded McjD is a result of the movement of TMs 1–2 and TMs 1′–2′ and we characterized this conformation by cysteine cross-linking L53C in the presence and absence of nucleotides, ATP/Mg$^{2+}$ and AMPPNP (Choudhury *et al*, 2014). Since the McjD-ADP-VO$_4$ structure is also in an outward-occluded conformation, this mutant was further characterized in the presence of ADP-VO$_4$. McjD-L53C is capable of forming cross-linked dimers in the presence of ADP-VO$_4$, suggesting that the high-energy post-hydrolysis McjD also adopts an occluded conformation in the *E. coli* membrane similar to our crystal structure (Figs 4A and EV3A).

Since an inward-occluded conformation without subunit intertwining has only been observed for McjD and not for other ABC exporters, we also characterized this new conformation by predictive cysteine cross-linking in ISOVs. We designed the A122C mutant at the beginning of TM3 on the cytoplasmic side of McjD, with a Cα-Cα′ distance of 7.3 Å. The McjD-A122C can form cross-linked dimers in ISOVs in the absence of nucleotides but less efficiently compared to the AMPPNP or ADP-VO$_4$ cross-linking experiments (Figs 4B and EV3B). This is the first time that an apo inward-occluded

conformation has been reported in *E. coli* membranes. A possible explanation for the reduced A122C cross-linking in ISOVs in the absence of nucleotides is that McjD alternates between inward-open and inward-occluded conformations. Nevertheless, the transporter is capable of adopting this conformation within the membrane. In the presence of MccJ25, we observed small enhancement of A122C cross-linking (Figs 4B and EV3C), suggesting that binding of the substrate brings the TM helices closer.

In the absence of nucleotides, the NBDs of ABC exporters disengage. There are conflicting reports regarding the degree of NBDs disengagement in bacterial ABC exporters (Borbat *et al*, 2007; Zou *et al*, 2009; Zoghbi *et al*, 2016). We sought to characterize the degree of McjD-apo NBD disengagement by cysteine cross-linking of the S509C mutant. This mutant was capable of forming cross-linked dimers even in the absence of CuCl$_2$, even after the ISOVs had been treated with 1 mM reducing agent, suggesting that the NBDs are also close within the *E. coli* membrane (Fig 4C). In the presence of AMPPNP and ADP-VO$_4$, the McjD-S509C can also form cross-linked dimers that are in agreement with our crystal structures (Figs 4C and EV3D). The predictive cysteine cross-linking data both in the presence and in the absence of nucleotides indicate that McjD mostly exists in an occluded conformation with its NBDs closely engaged in the *E. coli* membrane.

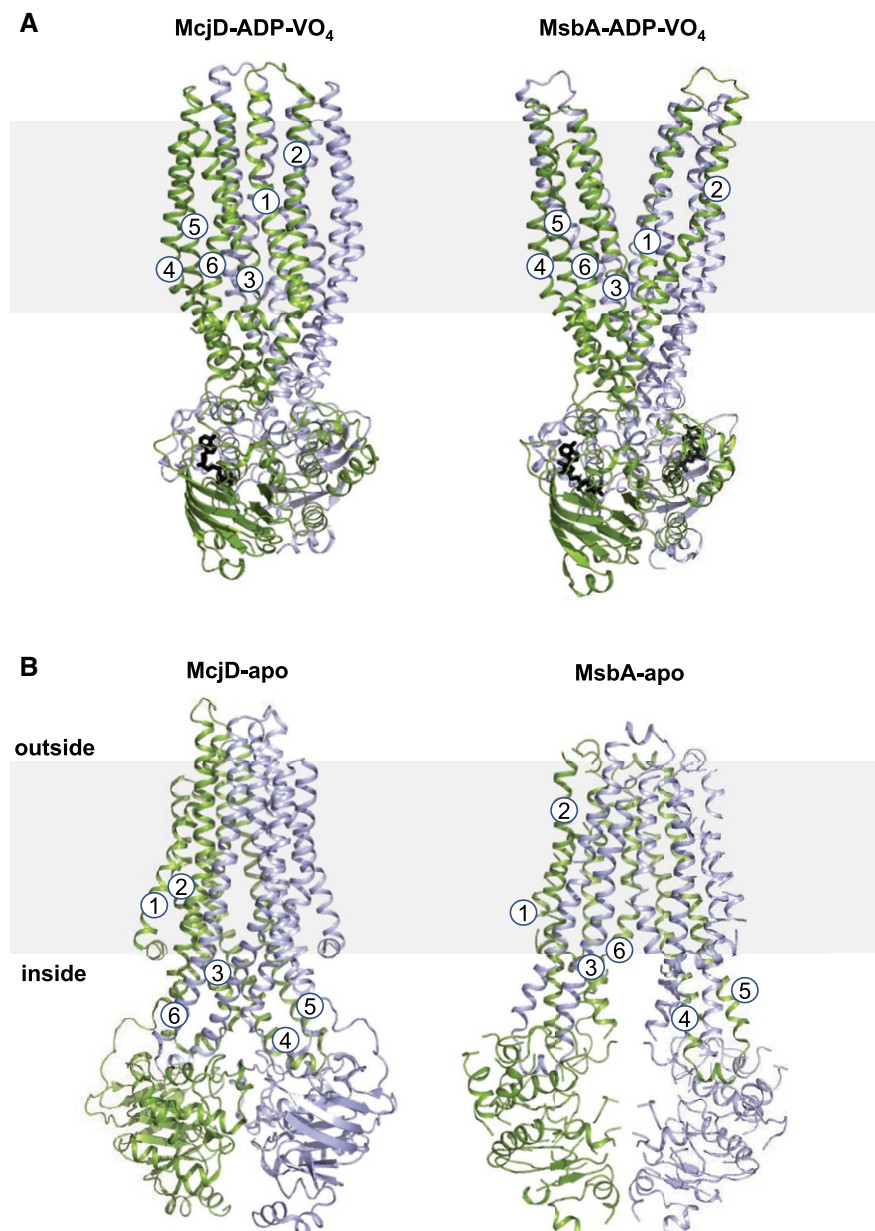

**Figure 2.  Structural comparison of McjD and MsbA.**

A   MsbA-ADP-VO$_4$ adopts an outward-open conformation with subunit intertwining (right panel), whereas McjD-ADP-VO$_4$ is in an occluded conformation (left panel).

B   MsbA-apo adopts an inward-open conformation with subunit intertwining (right panel). McjD-apo adopts an inward-occluded conformation as a result of movement of TMs 3–5 from one subunit towards the equivalent TMs of the opposite subunit, and subsequent loss of subunit intertwining (left panel).

Data information: MsbA and McjD are shown in cartoon and nucleotides in sticks. Each half transporter is coloured as in Fig 1. The transmembrane helices of one subunit are numbered. The membrane is depicted in grey.

## Pulsed electron-electron double resonance

Since our crystal structures and predictive cross-linking experiments revealed that McjD exists mostly in an occluded conformation, we performed Q-band PELDOR distance measurements on spin-labelled McjD mutants. We aimed to investigate the conformational changes associated with the export of the antibacterial peptide MccJ25 from the cytoplasmic to the periplasmic leaflet of the inner membrane. The data discussed below were recorded on McjD reconstituted in bicelles (Ward *et al*, 2014). Based on our crystal structures, a pair of R1 spin labels was introduced at the periplasmic side of the TMD at the loop connecting TMs 1 and 2 (L52R1), and another at the NBDs at position C547 (C547R1) since they could provide us with distances along the transport cycle and reveal stable states (Fig 5A). The ATPase activity of the labelled mutants was similar to the wild-type, unlabelled McjD. The

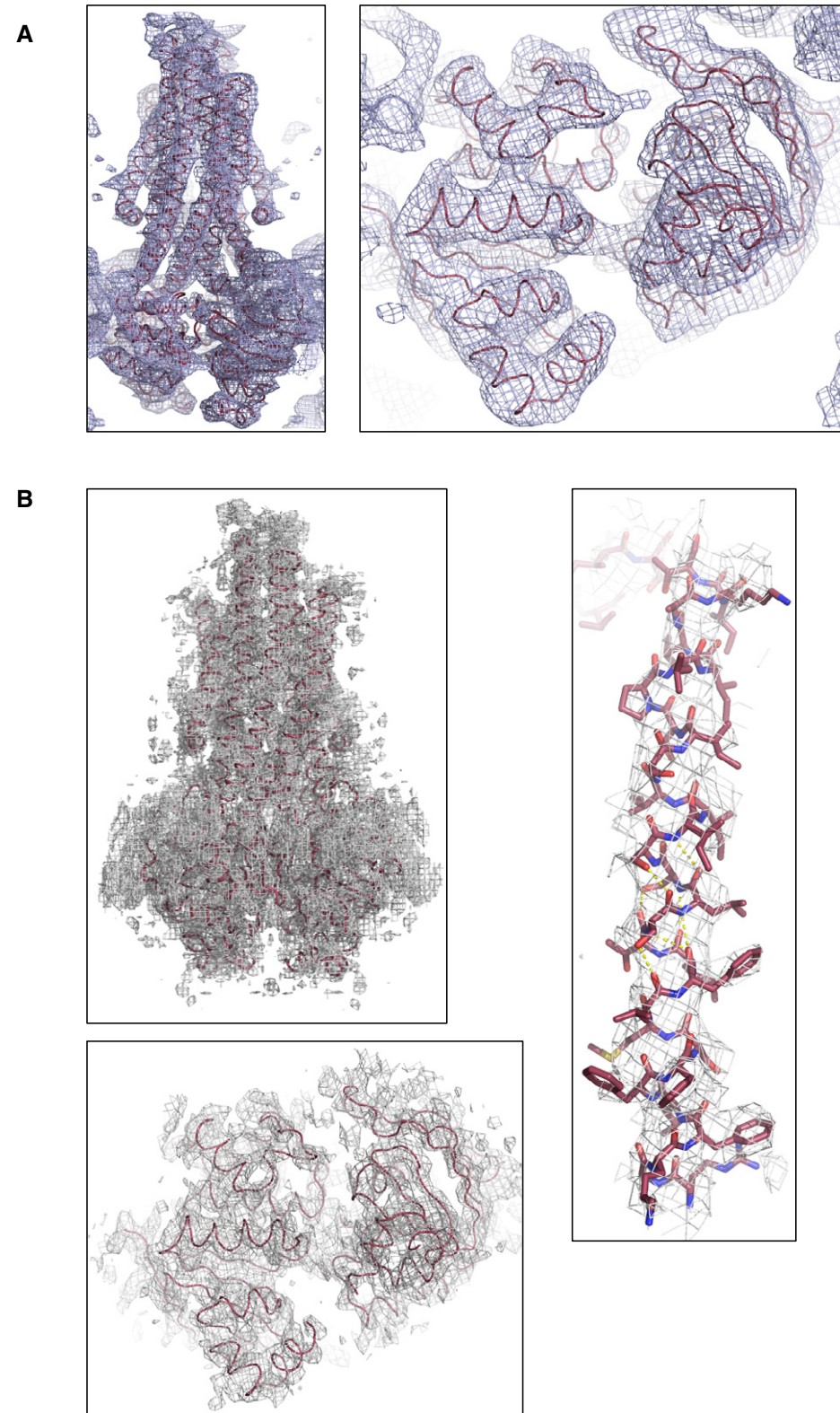

**Figure 3.  Electron density maps.**

A   Composite omit map (blue mesh) covering the McjD molecule (left panel). Right panel shows the composite omit map for the NBDs. Both maps are contoured at 1 σ.
B   Final 2|Fo|-|Fc| electron density map (grey mesh contoured at 1 σ) around McjD after refinement. Good-quality electron density could be observed around the TMD and NBDs. 2|Fo|-|Fc| electron density (grey mesh contoured at 1 σ) around TM 1 (right panel) and the NBDs after refinement (bottom panel). No electron density is present for nucleotides at the NBDs; the NBDs are in the same orientation as in panel (A).

    

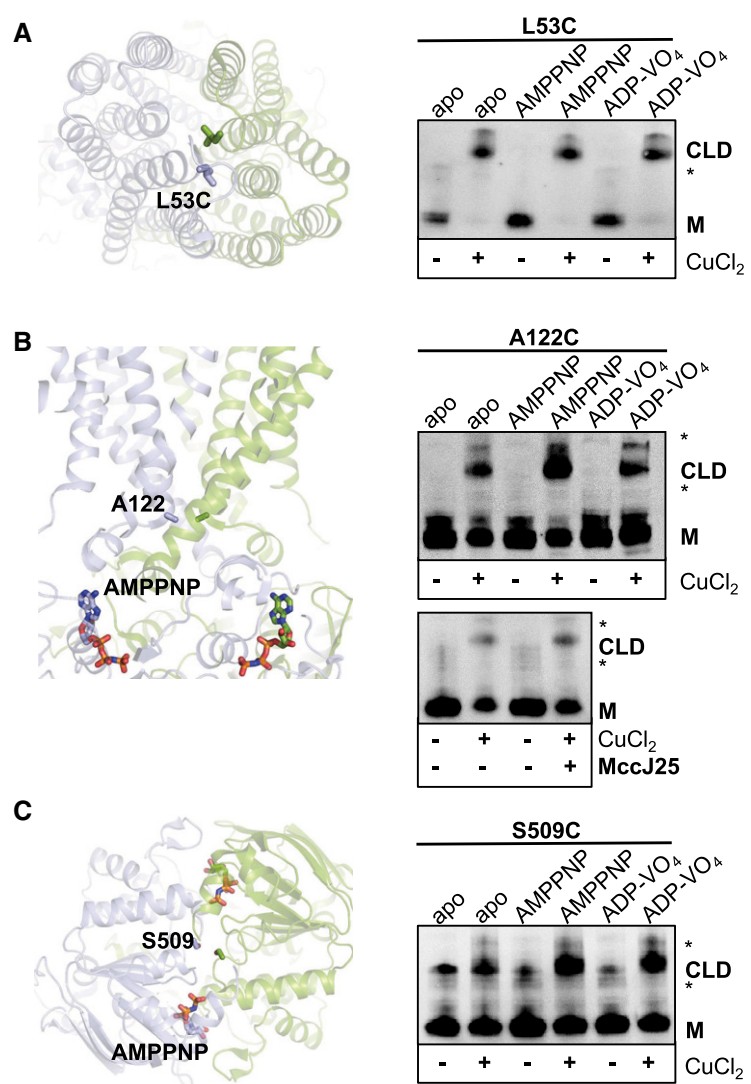

**Figure 4.  Predictive cysteine cross-linking along McjD in ISOVs.**

A   Cross-linking of L53C at the periplasmic side of the McjD TMD (left panel); L53 from each monomer is shown as sticks. L53C can be effectively cross-linked in the presence or absence of nucleotides (right panel).

B   Cross-linking of A122C at the cytoplasmic side of McjD (left panel); A122 from each monomer is shown as sticks. Cross-linking of the A122C is less effective in the absence of nucleotides compared to the nucleotide ones (right panel). MccJ25 induces small enhancement of cross-linking in the absence of nucleotides (bottom right panel and Fig EV3); the incubation with MccJ25 was performed in duplicate.

C   Cross-linking of S509C at the NBDs of McjD (left panel); S509 from each monomer is shown as sticks. Preformed cross-links can be observed even in the absence of CuCl$_2$ and DTT treatment, suggesting that the NBDs are in very close proximity (right panel). Inclusion of the nucleotides enhances the cross-linking dimer formation. The formation of the cross-linking dimers along the McjD transport cycle verifies that these conformations also exist in the *E. coli* inner membrane.

Data information: The reaction conditions for each lane are indicated above and below the gels (see Materials and Methods). All cross-linking experiments were visualized by Western blot. "CLD" denotes the formation of the cross-linking dimer and "M" the monomer in the absence of CuCl$_2$. Asterisk (*) denotes SDS-stable dimers. All gels contain the same amount of total membrane proteins and have not been normalized for McjD expression. Densitometry analysis of the gel band intensities are shown in Fig EV3.

Source data are available online for this figure.

mtsslWizard software (Hagelueken *et al*, 2012) was used to predict the interspin distances.

The L52R1 mutant produced PELDOR time traces of good quality with clearly visible oscillations. In the apo state, McjD L52R1 produces a sharp interatomic distance peak at 28 Å and a broad distribution of smaller peaks in the range between 40 and 60 Å (Fig 5B). The main peak at 28 Å is in close agreement with the

distance predicted from the crystal structures using the mtsslWizard software (Hagelueken *et al*, 2012; Fig 5B, grey shade).

It was previously shown that in the presence of AMPPNP or ADP-VO$_4$, the MsbA exporter alternates from an inward-open conformation to an outward-open conformation; PELDOR measurements of MsbA in the presence of ADP-VO$_4$ resulted in longer interatomic distances of 40 Å in both detergent and liposomes similar to

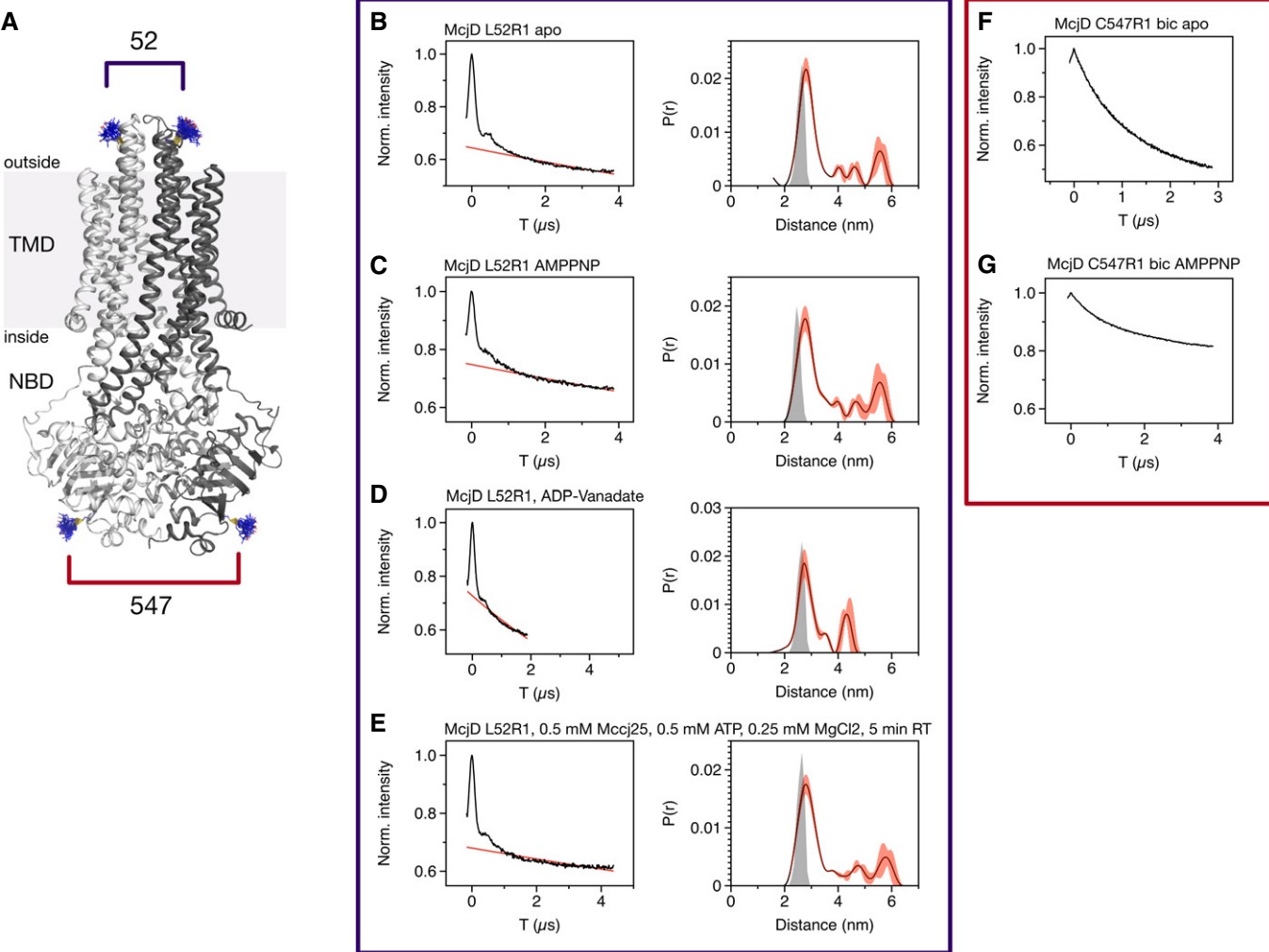

**Figure 5. PELDOR time traces and distance distributions of McjD L52R1 and 547R1 in bicelles.**

A    The McjD cartoon figure shows the position of the labelled side chains (blue lines) with respect to the structure.

B–G  For each measurement, the uncorrected PELDOR time trace is shown on the left as a black line and the fitted intermolecular background as a red line. On the right of time traces (B–E), L52R1, is the calculated distance distribution (black trace) with error bars as calculated by DeerAnalysis2016 (red). Grey shades are mtsslWizard predictions for the corresponding structure. The condition under which the time trace was recorded is indicated in each case.

the MsbA outward-open structure (Borbat *et al*, 2007). In contrast, our PELDOR data indicated that McjD did not undergo such large conformational changes upon nucleotide binding. The interatomic distance distributions in the presence of either AMPPNP or ADP-VO$_4$ were very similar to those of the apo protein (Fig 5C and D). This is in very good agreement with both our AMPPNP- and ADP-VO$_4$-bound crystal structures (Fig 5C and D, grey shades). We also investigated whether the presence of the substrate MccJ25 induces an outward-open conformation. McjD-L52R1 incubated with ATP/Mg$^{2+}$ and MccJ25 showed distance distributions very similar to those of the apo state (Fig 5D and E). A possible explanation for the apparent invariability of the interspin distances is that the outward-open conformation could be a short-lived and/or scarcely populated state, which would not be captured by PELDOR.

To analyse the movement of the NBDs of McjD during the transport cycle, we screened various cysteine mutants on the NBD surface. However, an active, well-behaved transporter could only be produced

when the native cysteine at position 547 was used as an anchor for the spin label. The corresponding PELDOR time traces are shown in Fig 5F and G. The time traces did not show any clear oscillations, indicating a highly dynamic structure at the spin-labelled position. As a consequence, it was not possible to unambiguously determine the intermolecular background and the underlying (presumably very broad) distance distributions cannot be reliably determined.

## Uptake of hoechst in proteoliposomes

Since we could not observe a stable outward-open McjD by PELDOR, we performed Hoechst transport assays in proteoliposomes using the L53C mutant to demonstrate that it transiently exists. We have previously shown that Hoechst 33342 (Hoechst) can be transported by McjD in proteoliposomes (Choudhury *et al*, 2014). In the absence of direct MccJ25 transport measurement in proteoliposomes, we reconstituted wild-type McjD in liposomes and

performed competition transport assays with Hoechst and MccJ25 (Fig 6A). In the absence of MccJ25 and presence of ATP, the proteoliposomes show reduced fluorescence for Hoechst, whereas in the absence of ATP, the fluorescence increased. In the presence of 10 molar excess MccJ25, we observe an increase in fluorescence relative to the ATP-only measurement, suggesting that MccJ25 competes with Hoechst for the McjD cavity and subsequent transport. L53C can be effectively cross-linked in the presence of $CuCl_2$ (Fig 4A) and our transport assays were performed in the presence and absence of the oxidizing agent to probe the opening of the periplasmic side of McjD (Fig 6B). In the absence of ATP, the proteoliposomes could not transport Hoechst, whereas the addition of ATP resulted in Hoechst uptake by McjD-L53C (Fig 6B). Incubation of L53C with $CuCl_2$, L53C$^{ox}$ and addition of ATP did not show any McjD-dependent Hoechst uptake (Fig 6B); thus, disulphide locking of the periplasmic side of McjD is detrimental to its transport activity. After 10 min of L53C$^{ox}$ "transport activity", the proteoliposomes were treated with 1,4-dithiothreitol (DTT), to reduce the disulphide bridges (L53C$^{re}$). McjD-L53C$^{re}$ was capable to mediate the ATP-dependent transport of Hoechst in proteoliposomes (Fig 6C and D) as a result of unlocking the periplasmic side of McjD, suggesting that McjD has to sample an outward-open conformation for the release of the substrate. L53C$^{re}$ shows a slightly reduced transport activity which could be due to incomplete reduction of all disulphides. This is our first evidence that McjD adopts an outward-open conformation that is mediated by both ATP and substrate.

## McjD cavity accessibility by PEGylation in ISOVs

Both our structural and functional data show that McjD exists mostly in an occluded conformation in the absence of a ligand. We have previously identified residues within the cavity that are involved in ligand recognition using ligand-induced ATPase assays and mutagenesis (Choudhury *et al*, 2014). Here, we probed the accessibility of the cavity by the substrate using cavity-specific cysteine mutants modified with methoxypolyethylene glycol maleimide 10,000 Da (mPEG10k) in ISOVs; we selected mPEG10k despite being significantly larger than MccJ25 (2.1 kDa), as it would enable us to observe a distinct electrophoretic mobility shift in SDS–PAGE. The mutants, N134C, I138C, I287C and T298C, were designed to study the cavity from its cytoplasmic to periplasmic side (Fig 7A). In the presence of 1 mM mPEG10k and absence of the nucleotides AMPPNP, ATP or ADP-VO$_4$, all mutants displayed higher molecular weight species (Fig 7B); some higher molecular weight species were also observed in the presence of nucleotides but these could be attributed to apo McjD (AMPPNP and ADP-VO$_4$ can be consumed by other ATP-binding proteins in the ISOVs; therefore, not all of the McjD molecules were inhibited). These assays were performed in triplicate with different membrane batches and provided similar results. mPEG10k is not membrane permeable and can only bind to solvent-accessible cysteines, suggesting that the McjD cavity becomes exposed to the aqueous solvent environment. This is the first evidence that McjD adopts an inward-open conformation that allows ligand access to its cavity. Owing to the homodimeric nature of McjD, the mutants are present in both halves of the transporter. The N134C and I138C mutants display stronger interaction with mPEG10k compared to I287C and T298C since they are located closer to the cytoplasmic side of the cavity (Fig 7A and B). The

I287C (I278C') and T298C (T298C') mutants are closer to the periplasmic side of the cavity and in closer proximity to each other; therefore, PEGylation of the first half transporter would prevent a second mPEK10k to bind to the equivalent site of the second half due to steric classes, which can explain the reduced PEGylation.

The only evidence that MccJ25 is recognized by McjD is by ligand-induced ATPase assays (Choudhury *et al*, 2014) and ligand binding studies by non-denaturing mass spectrometry (Mehmood *et al*, 2016). From these studies, it is unclear how the hydrophobic peptide MccJ25 enters the McjD cavity. We sought to answer the following question: (i) does MccJ25 enter through a lateral opening from inside the membrane as suggested for other ABC transporters (Aller *et al*, 2009), since McjD is found in an occluded conformation, or (ii) does it enter through an inward-open form of McjD? Therefore, we performed competition assays by pre-incubating McjD N134C, I138C and T298C mutant ISOVs with 1 mM MccJ25 followed by the addition of 1 mM mPEG10k. We have previously shown that MccJ25 cannot induce the ATPase activity of N134A due to the reduced affinity for MccJ25. Since N134C shows slightly reduced PEGylation in the presence of MccJ25, it is suggested that the cysteine mutation might also affect MccJ25 binding (Figs 7C and EV3E). I138C shows significantly reduced PEGylation, suggesting that it is involved in the binding of MccJ25 and protection from mPEG10k. In contrast, incubation of the T298C with MccJ25 resulted in abolishment of PEGylation of this site (Figs 7C and EV3E). T298C is found deep in the cavity and a bound MccJ25 would completely block access to mPEG10k. Our previous theoretical model of McjD with MccJ25 supports these data. Our competition data suggest that MccJ25 probably enters the inward-open McjD from the aqueous cytoplasmic side of the transporter rather than a lateral opening in the membrane. If McjD only existed in an occluded conformation and MccJ25 were to enter the cavity from the membrane, mPEG10k should not have been able to modify the N134C and T298C mutants since it is not membrane permeable and can only modify solvent-accessible cysteines. The sampling of an inward-open conformation is not affected by pH. The PEGylation is as effective at pH 9 as at pH 7.5 (Fig EV3F), suggesting that the pH does not induce an inward-occluded conformation.

## Molecular dynamics simulations

We used MD simulations to investigate whether changes in the outward-occluded state of the McjD transporter can be induced by the presence of ATP or ADP in the nucleotide binding sites, in the absence of the MccJ25 substrate. McjD-AMPPNP was chosen as input structure, and embedded in a POPC lipid bilayer (see Materials and Methods). The distance between specific pairs of residues in the periplasmic loops, in the cytosolic side of the cavity (cavity bottom) and in the NBDs was monitored during the simulation time.

### Periplasmic loops
L53 lies at the periplasmic end of TM1 of each monomer (Fig 8A). In the outward-occluded state, the Cα distance between the two L53 residues is 6 Å. In a previously modelled outward-facing state, the Cα distance between the two L53 pairs increases to 35 Å, as a consequence of the opening of the periplasmic end of the TM helices (Gu *et al*, 2015). During the simulation time, the L53 distance remains close to that of the input structure, in the presence of either ATP or

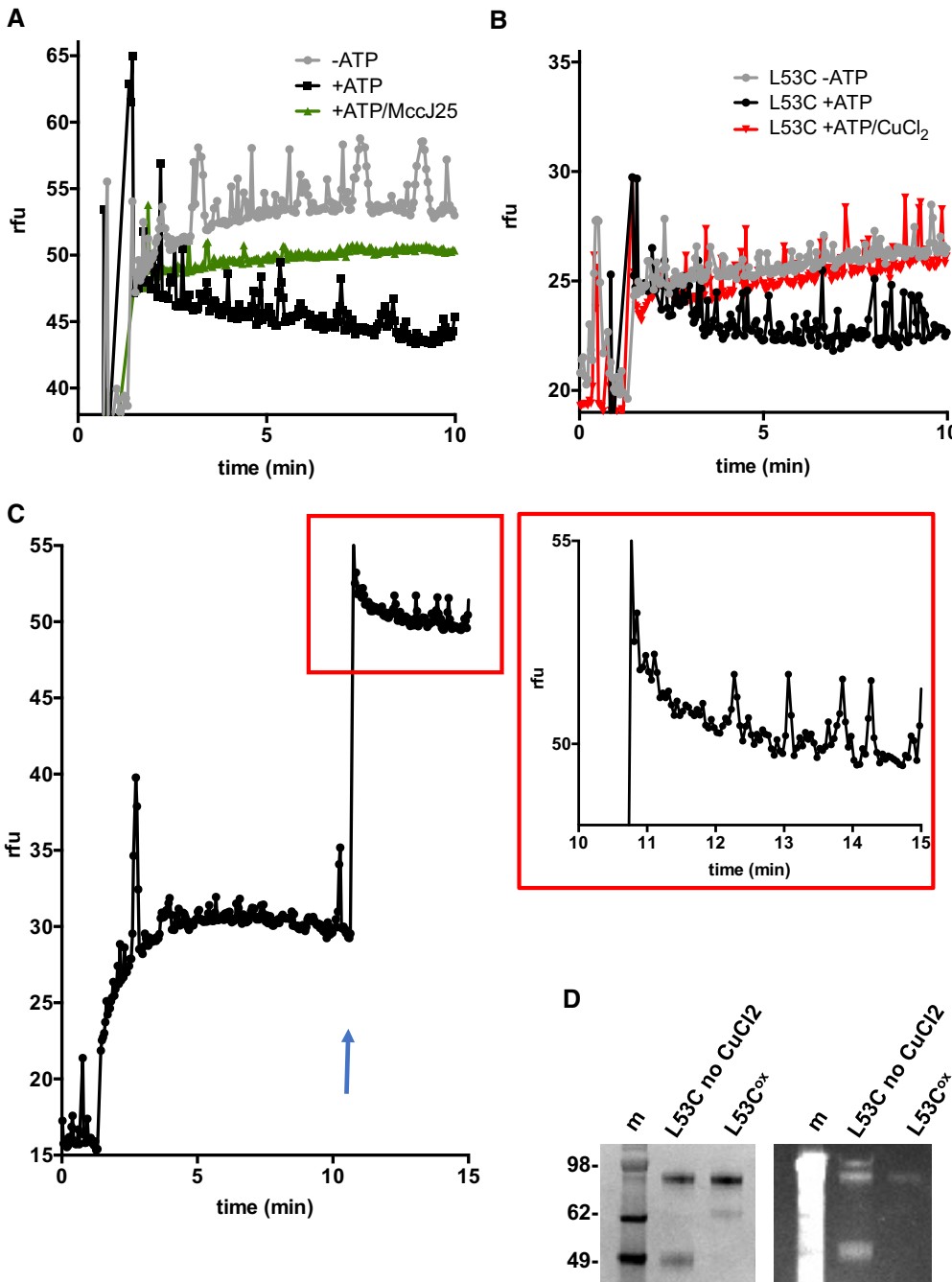

**Figure 6. Hoechst transport in proteoliposomes.**

A  McjD mediates the ATP-dependent uptake of Hoechst in proteoliposomes (black line); no uptake is observed in the absence of ATP (grey line). In the presence of MccJ25, an increase in Hoechst fluorescence is observed (green line).

B  McjD-L53C can transport Hoechst in proteoliposomes in the presence of ATP (black line), but not in the absence of ATP (grey line). L53C$^{ox}$ is not capable to transport Hoechst in the presence of ATP upon locking the periplasmic side of McjD (red line).

C  Upon reduction of the disulphides with DTT (at 10 min), McjD-L53C$^{re}$ was capable to uptake Hoechst. Blue arrow indicates the addition of DTT and ATP during the experiment; a spike in fluorescence is observed due to opening of the lid. The red boxed area is magnification of the McjD-L53C$^{re}$ Hoechst uptake.

D  The cross- and non-cross-linked proteoliposomes were analysed by SDS–PAGE to confirm the presence of disulphides. In the absence of CuCl$_2$, monomeric and SDS-stable dimeric McjD-L53C, lane 2, and mostly dimeric species for the L53C$^{ox}$, lane 3, were observed (left panel: Coomassie-stained). The effective formation of the cross-linked L53C$^{ox}$ was validated by 7-diethylamino-3-(4'-maleimidylphenyl)-4-methylcoumarin (CPM) dye that becomes fluorescent upon modification of free cysteines. The right panel shows a fluorescent imaged SDS–PAGE. L53C shows strong fluorescence, lane 2, whereas the L53C$^{ox}$ displays almost no fluorescence (a very small amount of fluorescence is present), lane 3. The molecular weight marker is shown in lane 1.

Data information: The curves/data points for panels (A–C) are the mean of two independent proteoliposome reconstitutions and transport measurements.
Source data are available online for this figure.

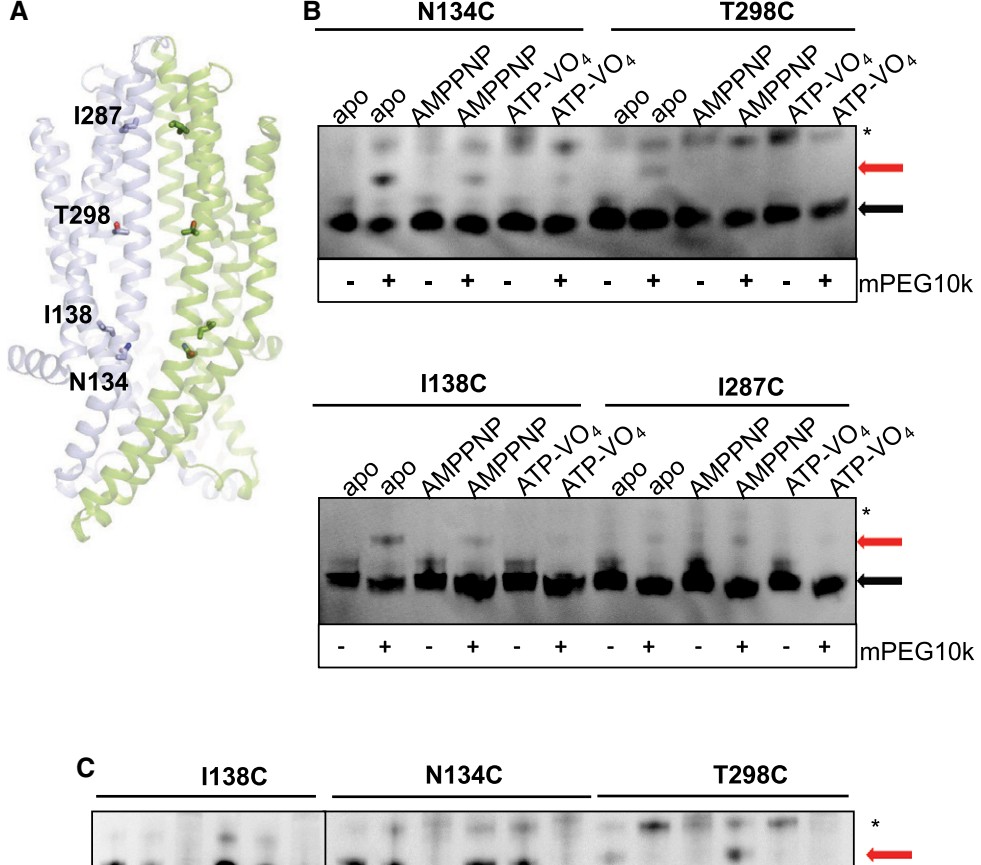

**Figure 7. PEGylation of the McjD cavity in ISOVs.**

A   The McjD TMD is shown in cartoon. The NBDs have been removed for clarity. The side chains of N134, I138, T298 and I287 that cover the length of the cavity are shown as sticks.

B   The cavity mutants could be effectively PEGylated by mPEG10k, suggesting that the transporter adopts an inward-open conformation. Some degree of PEGylation can also be observed in the presence of nucleotides but this is probably due to incomplete inhibition of McjD.

C   mPEG10k cannot PEGylate the T298C mutant in the presence of MccJ25, whereas I138C shows significantly reduced PEGylation in the presence of MccJ25. MccJ25 cannot compete with mPEG10k for the N134C mutant probably due to low affinity for the peptide. The MccJ25 competitions were performed in duplicate.

Data information: The reaction conditions for each lane are indicated above and below the gels (see Materials and Methods). All cross-linking experiments were visualized by Western blot. Red arrow denotes successful PEGylation, black arrow monomer and asterisk (*) SDS-stable dimers.

Source data are available online for this figure.

ADP, suggesting that no separation occurs between the TM1 helices (Fig 8A). To further assess possible structural changes at the periplasmic end of the TM helices, we monitored the distance between D51 (TM1) and S282 (TM6), highly conserved among the transporters used to generate the outward-facing model of McjD (Gu *et al*, 2015). In such a state, these residues move apart, and their Cα distance increases from ca. 8 of the outward-occluded state to ca. 25 Å. During the simulations, no opening motions are detected for these pairs of residues (Figs 8A and EV4A). However, a smaller intensity peak can be noticed for the L53–L53 pair at distances ranging from 12 to 14 Å, for both the ADP- and ATP-bound simulations (Figs 8A and EV4A). The structures with this larger separation,

taken after 600 ns, highlight a different arrangement for the L53–L53 pair (Fig EV4B). The opening between the two L53 residues is, however, still limited compared to the opening in the outward-facing state; thus overall, during the simulation time, we observe no transitions towards an outward-facing conformation of the periplasmic ends of the TMDs.

*Cavity bottom*

The Cα distance between the two A122 residues (Fig 8B) is 7 Å in the input structure. On average, this distance is ca. 6 Å for both the ADP- and the ATP-bound simulations (Fig 8B). For the ADP-bound systems, however, more simulations sample larger distances, up to

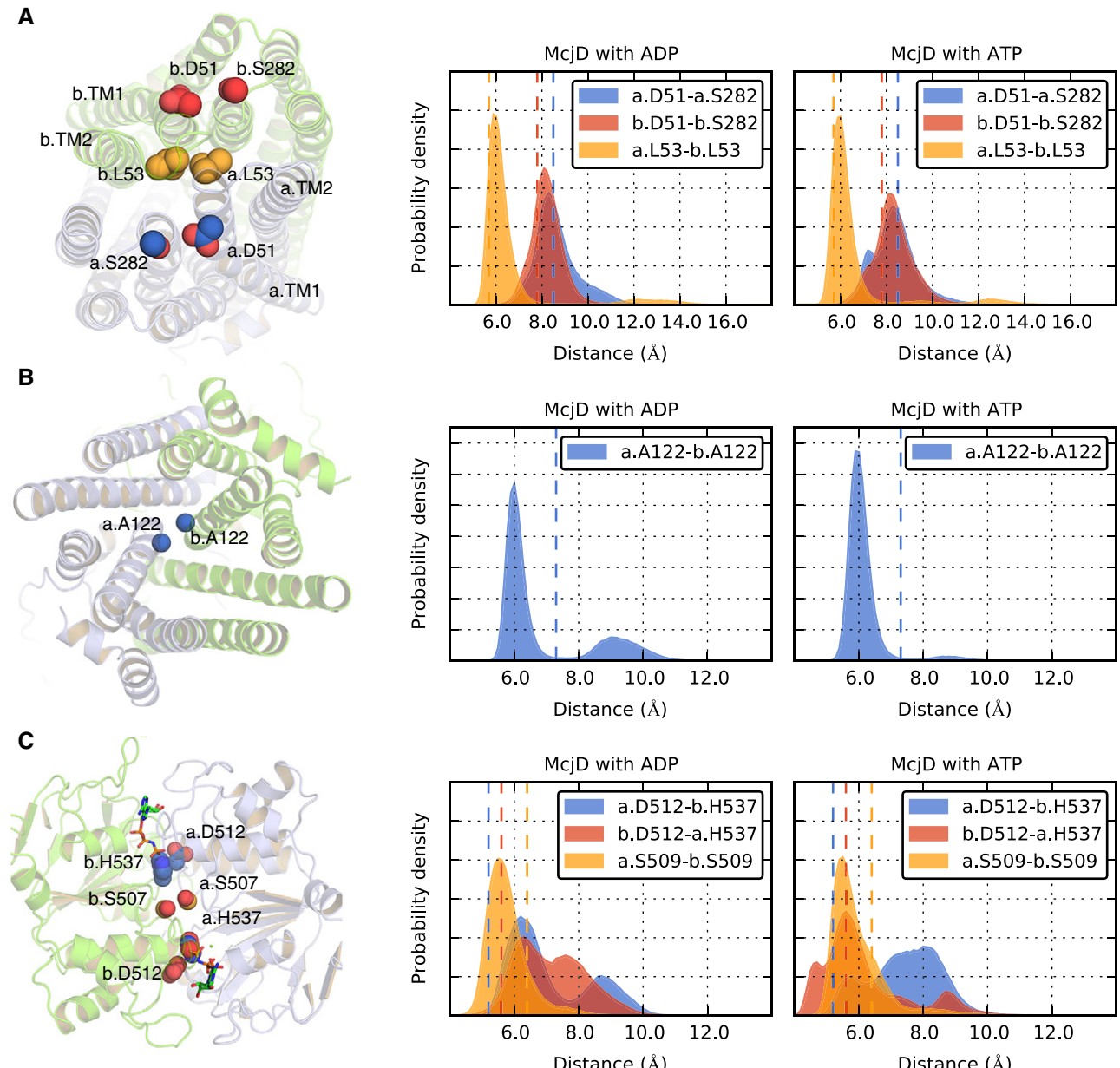

**Figure 8.   MD simulations of McjD.**

A–C   Distance distributions obtained from the ADP- and ATP-bound simulations for selected Cα pairs (A) in the periplasmic loops, (B) in the cytosolic side of the cavity and (C) at the NBD interface. The two McjD monomers are shown as light blue and green cartoons and labelled as "a" and "b", respectively. The side chains of the a.L53–b.L53, a.D51–a.S282 and b.D51–b.S282 pairs are shown in yellow, blue and red spheres, respectively. The dotted line in each graph shows the value of the corresponding distance in the starting structure.

ca. 12 Å (Fig EV4C), closer to the value of the outward-facing model (Gu *et al*, 2015).

### NBDs

The distance between the two S509 residues targeted by cross-linking studies is 6 Å in the initial structure and 8 Å in the apo McjD structure. During the simulation time, we observe no separation at the S509 level for both the ADP- and the ATP-bound systems (Figs 8C and EV4D). Limited NBD dissociation was also confirmed

by the distances between the two D512–H537 pairs, where D512 is the aspartate of the conserved D-loop and H537 is the histidine of the Switch motif (Figs 8C and EV4D).

## Discussion

In this study, we have determined the McjD structure in a novel conformation, apo inward-occluded and an additional

nucleotide-bound state, high-energy intermediate outward-occluded. These findings highlight novel mechanistic details on how McjD provides cells with self-immunity against antibacterial peptides. Since these new conformations have neither been previously reported nor observed for ABC transporters, we performed predictive cysteine cross-linking in *E. coli* ISOVs to verify their existence in the lipid environment. The cross-links covered the periplasmic side (L53), cytoplasmic side (A122) and NBDs (S509) of McjD, and they all formed effective cross-links along the transport cycle of McjD. To further study the occluded conformation of McjD, we carried out extensive MD simulations, starting from the AMPPNP-bound state, after replacing AMPPNP with either ADP or ATP molecules. These simulations, performed in the absence of the MccJ25 substrate, reveal no major opening of the transmembrane cavity at the periplasmic end of the TMDs (between the two TM1 helices based on the L53–L53 pair, and between TM1 and TM6 of each monomer, based on the D51–S282 pair), which would allow the transporter to switch towards an outward-facing state. The largest opening detected between the L53–L53 pair is of ca. 14 Å, significantly smaller than that of an outward-facing state or the minimum opening of ca. 20 Å (measured as distance between the centre of mass of the two L53 residues) required for allowing MccJ25 to be released on the periplasmic side of the membrane, as we have shown in previous simulation studies (Gu *et al*, 2015). The occlusion of the cavity resulted to be stable also at its cytoplasmic side (A122), with no evidence of a significant separation between the intracellular loops, in agreement with the cross-linking experiments. We acknowledge, however, that longer MD simulations might be required to explore the possibility that the larger L53–L53 distances detected in two simulations, together with the larger A122–A122 distances mainly sampled in some of the ADP-bound simulations, might trigger a conformational change towards an outward-facing state. Interestingly, an occluded state of the cavity was also sampled in previous MD simulations studies carried out starting from the outward-facing structure of Sav1866 in the presence of ADP (St-Pierre *et al*, 2012), or in the absence of nucleotide (Becker *et al*, 2010). Combined, our experimental and theoretical work provides strong evidence that McjD exists mostly in an occluded state within the membrane in the absence of the substrate MccJ25.

These data are in striking contrast to the previously published type I ABC transporter crystal structures and biochemical data that showed the existence of wide-open inward and outward conformations (Dawson & Locher, 2006; Borbat *et al*, 2007; Ward *et al*, 2007). In order to identify the mechanism that results in an outward-open McjD, we performed PELDOR measurements on McjD reconstituted in bicelles with spin-labelled L52C mutants, at the periplasmic side. Under all tested conditions, L52R1 revealed a distance distribution that fits the occluded conformation observed in the crystal structures. It is commonly thought that homodimeric multi-drug ABC transporters adopt an outward-open conformation upon nucleotide binding (Dawson & Locher, 2006; Ward *et al*, 2007). In contrast, heterodimeric ABC exporters assume this outward-open conformation upon ATP hydrolysis (in the form of ADP-VO$_4$; Mishra *et al*, 2014). McjD is the first report of a homodimeric ABC exporter that does not form a stable wide-open state on the periplasmic side in the presence of nucleotides. Interestingly, we could also not observe an opening of the periplasmic

side in the presence of both MccJ25 and ATP. Hence, we propose that opening of McjD is transient and not long-lived enough to be observed under our experimental conditions. Our transport data in proteoliposomes provide indirect evidence that McjD adopts an outward-open conformation; using the L53C mutant and in the presence of CuCl$_2$, McjD was not capable of transporting Hoechst compared to the untreated protein. Our data suggest that McjD does not need to adopt a stable or long-lived wide-open conformation to release its substrate. Indeed, our molecular simulations have previously shown that McjD requires an opening of ca. 20 Å to export MccJ25 from the cavity that reverts back to the occluded conformation soon after substrate release (Gu *et al*, 2015). Previous structural and biophysical data did not explain the mechanism of ATP-induced dimerization of the NBDs since they were found relatively far apart. Our cross-linking data show that the NBDs are in very close proximity to each other within the membrane; binding of ATP would be sufficient to allow dimerization of the NBDs without the need of large conformational changes. Previous MD studies carried out on the outward-facing state of Sav1866, where the NBDs are engaged in a tight dimer as in the outward-occluded state of McjD, showed different results, highlighting (i) partial NBD dissociation in the presence of ADP and inorganic phosphate (Oliveira *et al*, 2011); (ii) how the pattern of interactions between domains changes after hydrolysis of just one ATP molecule (Gyimesi *et al*, 2011) or after ATP removal (Becker *et al*, 2010); and (iii) the stability of this particular conformation even in the presence of ADP (St-Pierre *et al*, 2012). Here, our MD simulations, carried out at much larger timescales, reveal no dissociation at the NBD interface, independently by the presence of ADP or ATP. Our studies support the experimental findings that the binding of two nucleotide molecules at the NBDs (either ADP or ATP) is enough to establish a tight dimer coupled with the occluded conformation of the transmembrane cavity.

Finally, using PEGylation of the cavity and competition with MccJ25, we showed that MccJ25 is entering the cavity by the cytoplasmic side of the inward-open McjD rather than a lateral membrane opening as proposed for other ABC transporters (Aller *et al*, 2009). This is the first evidence that McjD adopts an inward-open conformation for peptide binding.

## Mechanism for antibacterial peptide export

ABC exporters alternate between inward- and outward-facing conformations to transport their substrates from the cytoplasm to the periplasm in an ATP-dependent manner (Beis, 2015). Structural and biochemical studies have shown that they adopt an inward-open conformation in the absence of nucleotides (Ward *et al*, 2007). In the presence of nucleotides, AMPPNP or ADP-VO$_4$, their NBDs dimerize and adopt an outward-open conformation. In both the inward- and outward-open states, ABC exporters display domain intertwining. The transition between the two states is not well understood. In our study, the nucleotide-bound occluded McjD did not display any domain intertwining and it was closed at both sides of the membrane that could explain this transition. McjD adopts conformations that are distinct from other structures and its mechanism cannot be explained with the current mechanistic model.

Therefore, taken together the biochemical, structural and simulation data we obtained allows us to propose a new working

hypothesis for the export of the antibacterial peptide MccJ25 (Fig 9), distinct from the mechanism established for other ABC exporters (Dawson & Locher, 2006; Ward *et al*, 2007). In the absence of nucleotides and substrate, McjD adopts an inward-occluded conformation that is sealed at both ends of the membrane and with the NBDs disengaged. Based on our PEGylation and predictive cross-linking data, McjD can also adopt an inward-open conformation for its cavity to become accessible to MccJ25. Binding of ATP alone results in the dimerization of the NBDs and formation of the nucleotide-bound outward-occluded conformation that does not induce conformational changes along the TMD. Futile ATP hydrolysis (in the absence of substrate) results in the restoration of the inward-occluded conformation that progresses via a high-energy transition state. While cells produce MccJ25, McjD is capable to sample an inward-open conformation for MccJ25 to enter the cavity. Binding of MccJ25 enhances the occlusion of the cavity as shown from our cross-linking data. Binding of ATP results in a transient outward-occluded conformation that is quickly followed by a transient/short-lived outward-open conformation that allows the release of the MccJ25 from the cavity. Release of the substrate results in the outward-occluded conformation. Finally, ATP hydrolysis resets the

transporter to an inward conformation to transport another MccJ25 molecule. We propose that both the binding of ATP and the substrate MccJ25 are necessary for McjD to open its periplasmic side for substrate release. The presence of the occluded conformations along the transport cycle probably acts as (i) a shielding mechanism for preventing peptide re-uptake by McjD and (ii) a mechanism to prime McjD for MccJ25 binding, during its biosynthesis, without building up toxic levels of the peptide (Choudhury *et al*, 2014). The structures of the peptidase-containing ATP-binding cassette transporter (PCAT) PCAT1 from *Clostridium thermocellum* (Lin *et al*, 2015) and the lipid-linked oligosaccharide flippase PglK from *Campylobacter jejuni* (Perez *et al*, 2015) have also been determined in a nucleotide-bound outward-occluded conformation and without domain intertwining similar to McjD. Therefore, the mechanism we propose for McjD is probably relevant to other ABC transporters that are dedicated to specific substrate export, whose cavity is "immediately shielded" upon substrate release. In contrast, the structure of the ABC transporter associated with antigen processing (TAP) does not display an occluded conformation (Oldham *et al*, 2016) and it has been shown that it is trans-inhibited by external peptide concentration, thus providing a mechanism to prevent the build-up of high

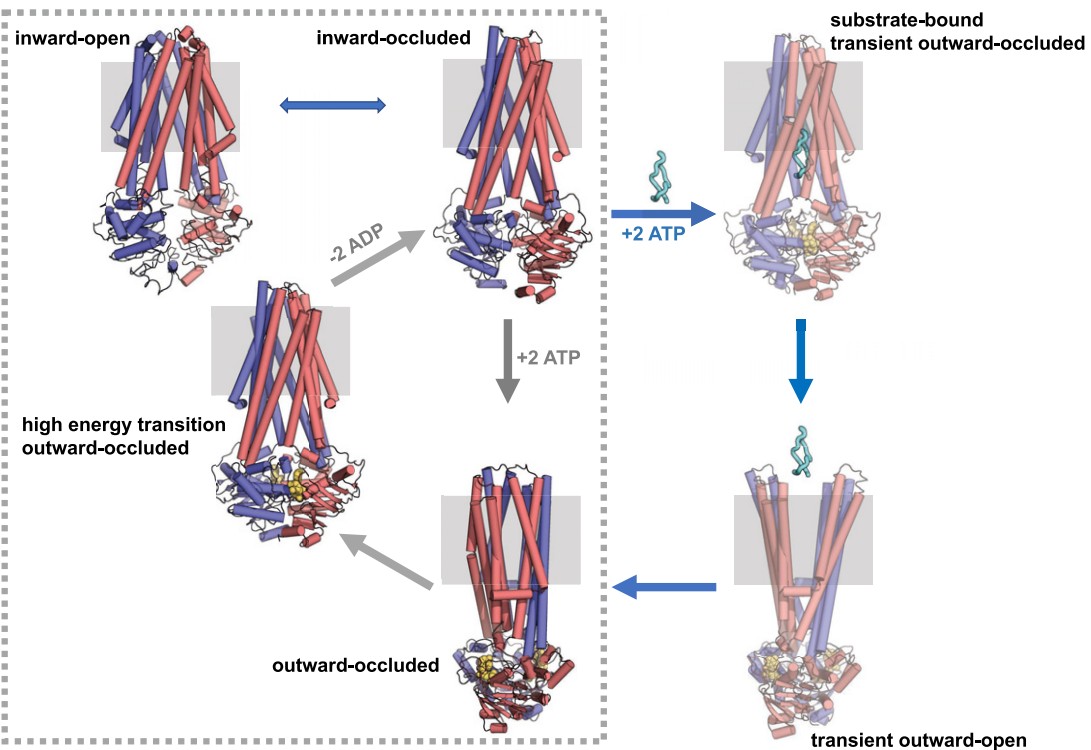

**Figure 9. Mechanism of antibacterial peptide export by McjD.**

McjD is shown in red and blue cartoons. The membrane is depicted in grey box. The nucleotides are shown as yellow spheres and MccJ25 as cyan ribbon. In the absence of nucleotides and peptide MccJ25, McjD adopts an inward-occluded conformation with its NBDs disengaged. In the presence of ATP and in the absence of MccJ25 (futile ATP hydrolysis), the NBDs dimerize and McjD adopts an outward-occluded conformation. ATP hydrolysis via a high-energy transition state resets the transporter to an inward-occluded conformation. The futile ATP hydrolysis cycle is boxed with broken lines. In the presence of MccJ25, McjD has to adopt an inward-open conformation to bind the peptide. Once it binds MccJ25 and ATP, it adopts a transient outward-occluded conformation that is quickly followed by a transient outward-open conformation that releases the peptide in the periplasmic side of the membrane. The two transient states are shown in light colour. Release of MccJ25 results in an outward-occluded conformation. The transporter resets to an inward-occluded conformation as in the futile cycle. The inward- and outward-open conformations of McjD are based on MsbA and Sav1866 structures.

levels of peptides in the ER that can induce stress (Grossmann *et al*, 2014). The distinct McjD conformations and mechanism described here can probably explain how ABC transporters, such as MsbA and Sav1866, have acquired multi-drug resistance in bacteria. We propose that shielding of the cavity is important to provide substrate specificity.

# Materials and Methods

## McjD purification and crystallization

McjD was purified in 0.03% dodecyl-maltopyranoside (DDM) as previously described without modifications (Choudhury *et al*, 2014). McjD-ADP-VO$_4$ was prepared by incubating 15 mg/ml McjD (20 mM Tris pH 7.8, 150 mM NaCl, 0.03% DDM) with 2 mM ATP, 2 mM sodium orthovanadate and 5 mM MgCl$_2$ at room temperature for 1 h. Crystals were grown at 293 K using the vapour diffusion method by mixing protein and reservoir solution at 1:1. Crystals were grown from a precipitant solution containing 10% PEG 4000, 0.1 M ammonium sulphate, 100 mM HEPES pH 7.5 and 22% glycerol. Crystals appeared overnight and they reached maximum size after 4 days. The crystals were directly frozen into liquid nitrogen, and diffraction screening and data collection were performed at Diamond Light Source synchrotron. These crystals diffracted X-rays to a maximum resolution of 2.9 Å.

McjD-apo crystals, 20 mg/ml McjD (20 mM Tris pH 7.8, 150 mM NaCl, 0.03% DDM), were grown from a precipitant solution containing 5 mM MgCl$_2$, 24% PEG400 and 100 mM glycine pH 9.0. Crystals appeared after a week and they reached maximum size after 2 weeks. Prior to freezing crystals, the cover slide was moved to a fresh reservoir solution containing 5 mM MgCl$_2$, 28% PEG400 and 100 mM glycine pH 9.0 and incubated overnight. Crystals were directly frozen into liquid nitrogen. Diffraction screening and data collection were performed at Diamond Light Source synchrotron. These crystals diffracted X-rays to a maximum resolution of 4.5 Å.

## Data collection

McjD-ADP-VO$_4$ diffraction data were collected at ~60 K on I23, the long wavelength Beamline at the Diamond Light Source (Wagner *et al*, 2016), close to the vanadium K-edge, λ = 2.26 Å. In total, 360° of data was recorded using the inverse-beam method (20° wedges). Images from a Pilatus 12 M detector were processed using autoPROC (Vonrhein *et al*, 2011) and STARANISO server (http://staraniso.globalphasing.org/cgi-bin/staraniso.cgi). Further processing was performed using the CCP4 suite (Collaborative Computational Project, 1994). Due to severe anisotropy, the resolution limit was restricted to 3.4 Å in the refinement stage. The space group was determined to be *C*2 with two copies of McjD in the asymmetric unit.

McjD-apo diffraction data to 4.7 Å were collected on I04-1 at Diamond Light Source at a wavelength of 0.92 Å using an ADSC Q315 detector and processed using xia2 (Winter, 2010). Further processing was performed using the CCP4 suite (Collaborative Computational Project N, 1994). The resolution of the data and anisotropy analysis were evaluated by half-dataset correlation

coefficient in Aimless (cut-off < 0.5; Evans, 2006). The space group was determined to be *P*4$_2$22 with one copy of McjD in the asymmetric unit.

The data collection statistics for both the McjD-ADP-VO$_4$ and apo crystals are summarized in Table 1.

## Structure solution and refinement

The McjD-ADP-VO$_4$ structure was determined by molecular replacement in Phaser (McCoy *et al*, 2007) using our previously published McjD structure (PDB ID: 4PL0; Choudhury *et al*, 2014) as search model. Initial refinement to 3.4 Å was carried out in REFMAC5 (Murshudov *et al*, 1997) and at later stages in Buster (Blanc *et al*, 2004). The structure was refined with restraints against the high-resolution structure 4PL0. After rigid body and restrained refinement, extra electron density corresponding to two ADP-VO$_4$ molecules and two MgCl$_2$ ions was identified (Fig EV1); these molecules were built and refined. To verify the presence of ADP-VO$_4$, anomalous difference maps were calculated in CCP4 using FFT. Strong electron density could be

**Table 1. Data collection and refinement statistics. Values in parentheses refer to data in the highest resolution shell.**

| Data collection statistics | apo | ADP-VO4 |
|---|---|---|
| Space group | *P*4$_2$22 | *C*2 |
| Resolution (Å) | 78.58–4.71 (4.83–4.71) | 77.3–3.40 (3.46–3.40) |
| Cell Dimensions (Å) | a = 87.8, b = 87.8, c = 351.5 | a = 235.3, b = 105.0, c = 117.4, β = 105.6° |
| Number of reflections | 28,417 (2025) | 143,753 (2852) |
| Number of unique reflections | 7,675 (534) | 22,352 (437) |
| Completeness (%) *spherical*[a] | 98.4 (94.4) | 58.6 (23.2) |
| Completeness (%) *ellipsoidal*[a] | | 94.2 (87.2) |
| Redundancy | 3.7 (3.8) | 6.4 (6.5) |
| R$_{merge}$ (%) | 8.2 (145) | 5.5 (63.5) |
| I/σ(I) | 9.6 (1.4) | 18.1 (2.6) |
| CC(1/2) | 0.99 (0.65) | 0.99 (0.86) |
| Refinement statistics | | |
| R$_{work}$ (%) | 31.4 | 25.7 |
| R$_{free}$ (%) | 33.4 | 26.0 |
| Average B-factors (Å$^2$) | 96.5 | 108 |
| Rms deviations from ideal values | | |
| Bonds (Å) | 0.01 | 0.01 |
| Angle (°) | 1.04 | 1.11 |
| Ramachandran plot outliers (%) | 1.23 | 0.7 |

[a]"spherical" looks at all data within a specific, spherical resolution range/bin, so this is the usual, well-known way of looking at data as a function of resolution. "Ellipsoidal" additionally requires that a data point be within the fitted ellipsoid in order to be considered.

observed up to 10 σ at the vanadium position at both NBDs. Due to the medium resolution of the data, placement of the $VO_4$ was guided by the anomalous difference maps. The final model has an $R_{work}$ of 25.7% and an $R_{free}$ of 26.0%. The McjD-ADP-$VO_4$ structure has 93.3% of the residues in the favoured Ramachandran region and has eight outliers as calculated by MolProbity (Chen *et al*, 2010).

Attempts to determine the structure of McjD-apo using the full-length McjD monomer failed in both Phaser and MOLREP (Vagin & Teplyakov, 1997); therefore, the structure was split in two search domains, TMD and NBD. Phaser under Phenix (Adams *et al*, 2002) was able to locate the TMD and NBD with good packing scores. Similar solution could also be obtained using MOLREP in CCP4. Refinement to 4.7 Å was carried out in Buster (Blanc *et al*, 2004). Good electron density could be observed for both the TMD and NBDs including density for aromatic side chains (Fig 3). Continuous electron density could also be observed for the linker that connects the TMD with the NBD. Due to the low resolution, we calculated composite omit maps in CCP4 to remove any model bias from molecular replacement (Fig 3). Overall B-factors were refined with the final model having an $R_{work}$ of 31.4% and an $R_{free}$ of 33.4%. The McjD-apo structure has 89.5% of the residues in the favoured Ramachandran region and has seven outliers as calculated by MolProbity (Chen *et al*, 2010). All model building was performed in Coot (Emsley & Cowtan, 2004).

The refinement statistics for both structures are summarized in Table 1. The coordinates and structure factors of McjD-ADP-$VO_4$ and McjD-apo have been deposited to the Protein Data Bank with PDB ID codes 5OFR and 5OFP, respectively.

### MccJ25 purification

MccJ25 was produced from cultures of *E. coli* K12 MC4100 harbouring the plasmid pTUC202 and purified as previously described (Zirah *et al*, 2011).

### Site-directed mutagenesis

McjD mutants were generated using the QuickChange Site-Directed Mutagenesis Kit (Agilent Technologies). DNA sequencing of the resulting plasmids confirmed the presence of the mutants.

### Cysteine cross-linking

Cysteine cross-linking experiments in ISOVs were performed as previously described without modifications (Choudhury *et al*, 2014), with the exception of S509C mutant that was pre-treated with 1 mM DTT for 30 min at room temperature to reduce preformed cysteine cross-links. The effect of the MccJ25 to induce cysteine cross-links of A122C was determined by adding 1 mM MccJ25 to A122C ISOVs for 10 min prior to $CuCl_2$ addition. Formation of cysteine cross-links was analysed by Western blotting. The gels were imaged using an ImageQuant LAS4000 (GE Healthcare). Densitometry analysis of the gel bands intensities was performed using the ImageQuant TL software (GE Healthcare); to quantify the extent of MccJ25-stimulated cross-linking of A122C, band intensities (automatic peak detection mode) corresponding to CLD formation were estimated for each gel lane.

### Hoechst 33342 transport in proteoliposomes

McjD and McjD-L53C were reconstituted in proteoliposomes using the rapid dilution protocol. In brief, *E. coli* polar lipids were made to 20 mg/ml in water and were bath-sonicated for 15 min until the solution became less hazy. 100 μl of 0.1 mg/ml McjD or McjD-L53C was mixed with 20 μl lipid stock on ice for 10 min. The solution was further diluted into 4 ml of liposome buffer (50 mM Tris pH 8 and 50 mM KCl) and was incubated for further 5 min on ice. Proteoliposomes were pelleted by centrifugation at 45,000 *g* for 30 min. The supernatant was carefully removed and the pellet was resuspended in 100 μl liposome buffer. The proteoliposomes were immediately used in experiments. Uptake measurements were performed as before with minor modifications (Choudhury *et al*, 2014); the amount of protein reconstituted in liposomes was 0.4 mg/ml and the final Hoechst concentration was 0.1 μM. Data were recorded using a Cary Eclipse Fluorescence Spectrophotometer (Agilent Technologies). The mean value of two independent experiments was calculated and plotted in GraphPad Prism.

### PEGylation

Inside-out vesicles of each cavity mutant were prepared as for the cysteine cross-linking experiments. The concentration of the ISOVs was adjusted to a total protein concentration of 10 μg/μl in 50 mM HEPES pH 7.5 and 150 mM NaCl. Where appropriate, ISOVs were incubated with 3 mM nucleotides and $MgCl_2$ at 37°C for 20 min. 1 mM mPEG10k was then added to the relevant conditions and incubated at room temperature for 1 h. PEGylation reactions were performed in triplicate. Competition assays were performed in duplicate at a final concentration of 1 mM MccJ25 (in DMSO) for 1 h with an equimolar ratio of mPEG10k. All reactions were quenched upon the addition of 2 mM *N*-ethylmaleimide and ran on nonreducing NuPAGE Bis-Tris gels. PEGylation results were analysed by Western blotting. The gels were imaged using an ImageQuant LAS4000 (GE Healthcare). Densitometry analysis of the gel band intensities was performed using the ImageQuant TL software (GE Healthcare); the degree of PEGylated species formed was quantified for the competition assays using the automatic peak detection setting.

### Labelling of McjD mutants for PELDOR measurements

PELDOR McjD mutants were purified as the wild-type McjD with minor modifications to account for the labelling. In brief, McjD mutants in DDM after the reverse chromatography step (Choudhury *et al*, 2014) were concentrated to 500 μl using a 100-kDa MW cut-off concentrator. The protein was incubated with 10-fold molar excess of (1-oxyl-2,2,5,5-tetramethyl-Δ3-pyrroline-3-methyl) methanethiosulfonate dye (Toronto Research Chemicals) dissolved in DMSO. The solution was placed on an elliptical roller in dark for 3 hours. Excess label was removed by passing the protein through a Superdex S200 10/300 GL (GE Healthcare) equilibrated in 0.06% DDM, 300 mM NaCl and 100 mM TES pH 7.5, made in $D_2O$. Fractions were monitored using an ÄKTA FPLC (GE Healthcare) at an absorbance of 280 nm. McjD fractions were concentrated using a 100-kDa MW cut-off concentrator, until a protein concentration of 18 mg/ml was reached. The sample was flash-frozen in liquid $N_2$. *cw*-X-band EPR spectra of the samples were recorded and are shown in Fig EV5. The

                                                       

spectra show the typical broadening of the nitroxide spectrum that is due to the immobilization of the nitroxide on the protein. Note that small amounts of free label are visible in the spectra.

## ATPase activity of purified McjD and mutants

The ATPase activity of the wild-type McjD-, McjD-L53C- and PELDOR-labelled McjD mutants was measured as previously described (Choudhury *et al*, 2014).

## PELDOR measurements

For PELDOR measurements, the spin-labelled McjD samples were reconstituted into bicelles as previously described (Ward *et al*, 2014). If needed, the samples were supplemented with nucleotides or substrate and incubated at room temperature. The samples were supplemented with 50% deuterated ethylene glycol and transferred to a 3-mm quartz Q-band EPR tube and flash-cooled in liquid nitrogen. The PELDOR time traces were recorded on a Bruker ELEXSYS E580 pulsed Q-band EPR spectrometer, with an ER 5106QT-2 Q-band resonator. The instrument was equipped with a continuous flow helium cryostat (CF935) and temperature control system (ITC 502), both from Oxford instruments. The second microwave frequency was coupled into the microwave bridge using a commercially available setup from Bruker. All pulses were amplified via a 150-W pulsed travelling wave tube (TWT) amplifier. PELDOR experiments were performed with the pulse sequence $\pi/2(\nu_A)\text{-}\tau_1\text{-}\pi(\nu_A)\text{-}(\tau_1 + t)\text{-}\pi(\nu_B)\text{-}(\tau_2 - t)\text{-}\pi(\nu_A)\text{-}\tau_2\text{-}echo$. The detection pulses ($\nu_A$) were set to 12 ns for the $\pi/2$ and 24 ns for the $\pi$ pulses and applied at a frequency 80 MHz lower than the resonance frequency of the resonator. The pulse amplitudes were chosen to optimize the refocused echo. The $\pi/2$-pulse was phase-cycled to eliminate receiver offsets. The pump pulse ($\nu_B$) was set at the resonance frequency of the resonator and its optimal length (typically 16 ns) was determined using a transient nutation experiment for each sample. The field was adjusted such that the pump pulse is applied to the maximum of the nitroxide spectrum. The pulse amplitude was optimized to maximize the inversion of a Hahn-echo at the pump frequency. All PELDOR spectra were recorded at 50 K with an experiment repetition time of 1 ms, a video amplifier bandwidth of 20 MHz and an amplifier gain of 42 dB. $\tau_1$ was set to 260 ns and the maximum of $\tau_2$ was set to values ranging from 2 to 6 $\mu$s. Deuterium modulation was suppressed by the addition of eight spectra of variable $\tau_1$ with a $\Delta\tau_1$ of 16 ns. The obtained time traces were divided by a monoexponential decay to eliminate intermolecular contributions and renormalized. Distance distributions were obtained from the background-corrected data by using the Tikhonov regularization as implemented in DeerAnalysis2016 developed by Gunnar Jeschke (Jeschke *et al*, 2006). The influence of different starting points for the background fitting was analysed with the evaluation feature of DeerAnalysis. The PyMOL (www.pymol.org) plugin mtsslWizard (Hagelueken *et al*, 2012) was used to predict distance distributions.

## Molecular dynamics simulations

MD simulations were performed on the McjD-AMPPNP structure, obtained at a resolution of 2.7 Å (PDB ID: 4PL0; Choudhury *et al*, 2014). Two sets of simulations were built by replacing the AMPPNP molecule with either ATP or ADP, while the position of the $Mg^{2+}$ ions was kept as in the crystal structure. H537, the Switch motif, was set protonated for the ATP-bound simulations, and neutral for the ADP-bound ones. The protein in complex with either ATP or ADP was embedded in a 1-palmitoyl-2-oleoyl-sn-glycero-3-phosphocholine (POPC) bilayer using g_membed (Wolf *et al*, 2010), with a total of 500 lipids. The simulation box, including water and ions to neutralize the system, was $13 \times 13 \times 16$ nm. Energy minimization, equilibration protocols and production runs were carried out with GROMACS 4.5.5 (Berendsen *et al*, 1995; Hess *et al*, 2008), using the ff54a7 version of the GROMOS 96 force field (Poger & Mark, 2010; Poger *et al*, 2010; Schmid *et al*, 2011).

The ATP-bound system was energy-minimized with (1) position restraints on the protein heavy atoms (force constant of 1,000), ADP/ATP molecules and $Mg^{2+}$ ions (force constant of 10,000 kJmol$^{-1}$ nm$^{-2}$) first; (2) then, with position restrains only on the backbone atoms (force constant of 10,000 kJmol$^{-1}$ nm$^{-2}$). For the ADP-bound system, we performed an additional intermediate step between (1) and (2), where we applied position restraints on the protein backbone atoms, ADP/ATP molecules and $Mg^{2+}$ ions (force constant of 10,000 kJmol$^{-1}$ nm$^{-2}$), before the final energy minimization.

After energy minimization, the ATP-bound system was initially equilibrated for 10 ns, during which the protein heavy atoms were position-restrained with a force constant of 1,000 kJmol$^{-1}$ nm$^{-2}$, while ATP/ADP molecules and $Mg^{2+}$ ions were restrained with a 10,000 kJmol$^{-1}$ nm$^{-2}$ force constant. 5 additional ns was performed with position restraints on the protein backbone atoms (force constant of 10,000 kJmol$^{-1}$ nm$^{-2}$). For the ADP-bound system, this step was preceded with 5 ns carried out with position restraints applied on the protein backbone atoms and on the ADP molecules and $Mg^{2+}$ ions as well (force constant of 10,000 kJmol$^{-1}$ nm$^{-2}$). A time step of 2 fs, the Berendsen barostat (Berendsen *et al*, 1984; with a relaxation time constant of 2 ps and a semi-isotropic reference pressure of 1 bar) and the V-rescale thermostat (Bussi *et al*, 2007; with a time constant of 0.1 ps and a reference temperature of 313.15 K) were used for the equilibration steps.

The two ADP- and ATP-bound McjD structures resulting from the equilibration protocol were used as starting structures for production runs. For the ADP-bound system, 10 independent runs, of 600 ns each, were generated. For the ATP-bound system, 10 independent runs were also performed, eight of 600 ns each and two more of 580 and 440 ns, respectively. A time step of 2 fs, the Parrinello–Rahman barostat (Parrinello & Rahman, 1981; with a relaxation time constant of 6 ps and a semi-isotropic reference pressure of 1 bar for coupling) and the V-rescale thermostat (Bussi *et al*, 2007; with a relaxation time constant of 2.0 ps and a reference temperature of 313.15 K) were used for the production runs.

Distances between the selected residues described in the Results were calculated as C$\alpha$ distances. The probability density of the distances shown in the Results section is the average probability density obtained from the 10 simulations of each set (ADP- or ATP-bound). Plots were generated with Matplotlib (Hunter, 2007), while protein figures, unless otherwise specified, were made with The PyMOL Molecular Graphics System, version 1.7 Schrödinger, LLC (DeLano, 2002).

**Expanded View** for this article is available online.

## Acknowledgements

We would like to thank Diamond Light Source for beam time allocation and access, and Dr Marc Morgan, Imperial College London, for help with scheduling. We would like to thank Dr Gerard Bricogne, Dr Clemens Vonrhein and Dr Ian Tickle for help with the anisotropic correction of the data. We would like to thank Prof Bob Ford and Prof Alex Cameron for critical feedback on the manuscript. KBo is supported by a Biotechnology and Biological Sciences Research Council (BBSRC) Doctoral Training Partnership (DTP) Studentship. WYW is supported by a Mobility Starting Grant from Swedish Research Council FORMAS. DPT is an Alberta Innovates Health Solutions Scientist and Alberta Innovates Technology Futures Strategic Chair in BioMolecular Simulation. DPT is supported by the Canadian Institutes for Health Research (MOP-62690). MD simulations were carried out on WestGrid/Compute Canada facilities. This work was supported by the Medical Research Council (MR/N020103/1 to KBe).

## Author contributions

KBe designed and managed the overall project. KBo, HGC, IM, KEO, AW and KBe grew crystals, collected data, built and refined the structures. KBo and WYW performed PEGylation, cross-linking experiments and transport assays. KBo, WYW and HGC prepared proteins for PELDOR measurements. GH and OS performed PELDOR measurements and analysis. SZ and SR produced and purified the MccJ25. VC and DPT performed and analysed molecular simulations. KBe wrote the manuscript with help from the other authors.

## Conflict of interest

The authors declare that they have no conflict of interest.

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
