## [Review Process File · The EMBO Journal]

Manuscript EMBO-2017-97278

Structural basis for antibacterial peptide self-immunity by the bacterial ABC transporter McjD

Kiran Bountra, Gregor Hagelueken, Hassanul G. Choudhury, Valentina Corradi, Kamel El Omari, Armin Wagner, Indran Mathavan, Séverine Zirah, Weixiao Yuan Wahlgren, D. Peter Tieleman, Olav Schieman, Sylvie Rebuffat, Konstantinos Beis

Corresponding author: Konstantinos Beis, Imperial College London

Review timeline:	Submission date:	10 May 2017
	Editorial Decision:	12 June 2017
	Revision received:	13 July 2017
	Editorial Decision:	27 July 2017
	Revision received:	03 August 2017
	Accepted:	09 August 2017

Editor: Ieva Gailite

Transaction Report:

1st Editorial Decision	12 June 2017
--------------

Thank you for submitting your manuscript for consideration by the EMBO Journal. We have now received three referee reports on your manuscript, which I am copying below for your information.

As you can see from the comments, all three referees express interest in the work and the topic. However, they also raise several concerns that need to be addressed in order to consider publication here. Given the referees' positive recommendations, I would like to invite you to submit a revised version of your manuscript in which you address the comments of all three referees.

I should add that it is The EMBO Journal policy to allow only a single major round of revision and that it is therefore important to resolve the main concerns raised at this stage.

When preparing your letter of response to the referees' comments, please bear in mind that this will form part of the Review Process File, and will therefore be available online to the community. For more details on our Transparent Editorial Process, please visit our website: http://emboj.emboress.org/about#Transparent_Process

We generally allow three months as standard revision time. Please contact us in advance if you would need an additional extension. As a matter of policy, competing manuscripts published during this period will not negatively impact on our assessment of the conceptual advance presented by your study. However, we request that you contact the editor as soon as possible upon publication of any related work to discuss how to proceed.

Please feel free to contact me if have any further questions regarding the revision. Thank you for the opportunity to consider your work for publication. I look forward to your revision.

REFEREE REPORTS

Referee #1:

The authors of this study have investigated the transport mechanism of the prokaryotic peptide ABC exporter McjD. By using a range of techniques that include X-ray crystallography, cross-linking, EPR spectroscopy and computer simulations they have attempted to capture different states of the transporter and relate them to intermediates of the transport cycle. Described experiments and simulations have been carefully carried out and the presented data appears to be of high quality and to a large degree conclusive.

McjD was crystallized from distinct conditions with the aim to lock the transporter in different states. The structure in complex with ADP and VO4 (mimicking the transition state of ATP hydrolysis) is very similar to a previous structure with the non-hydrolysable ATP analogue AMP-PNP. The presence of both nucleotide and VO4 in the binding site is convincingly demonstrated. A second structure of the protein in the absence of nucleotides, which was determined at lower resolution, shows a rearrangement of the nucleotide binding domains but no appreciable differences in the transmembrane domain. Since, under equivalent conditions, other ABC exporter structures show a large conformational transition into an inward-facing conformation, the authors conclude that the observed occluded state of the transporter is stable and inward- and outward-facing conformations in McjD might only be transiently occupied. The fact that detergent-solubilized membrane transport proteins reside in a single conformation in a crystalline environment is not uncommon and does on its own not necessarily allow conclusions on other conformations of the transport cycle. However, the authors provide ample of evidence from complementary studies, which suggest that their claim of a predominance of the occluded conformation during transport with transient transitions into outward- and inward-facing states may be a valid. These studies include cross-linking of the transporter in different states, EPR experiments, accessibility studies to residues of the substrate binding site and transport assays in proteoliposomes. In latter, the introduction of a cross-link at the extracellular side of the transmembrane domain prevents transport in a reversible manner by preventing the transition into an outward-facing state. The conclusion on the conformational stability of the observed McjD structure is also supported by Molecular Dynamics simulations, which show little changes in different conditions.

In light of their attempts to study McjD in different conformations, the described results might at first glance appear unspectacular but they build a credible case about peculiarities in the transport mechanism of this peptide transporter that may differ from other ABC exporters. In summary I think that the authors have provided a large amount of interesting data and they have done a good job in their interpretation and thus think that the described work is a strong candidate for publication in the EMBO Journal.

I have several comments:

- In the pictures it is not clear how the ADP-VO4 and apo-structures differ. A stereo figure of a superposition of CA traces would help the reader to get an impression.
- The term outward- and inward-facing in an alternate access transporter usually refers to a distinct conformation of the transmembrane domain where a cavity leading to the substrate binding side is either oriented towards the outside or inside. Along these lines in an outward- or inward-occluded conformation, the binding site might not be accessible but the conformation should still contain features of and outward- or inward-facing states. Since the transmembrane domain is the same in all of the three known structures the authors might want to reconsider their terminology.
- The authors use the term transition state for one of their structures (page 5 line 2). Since such transition state is on an energy maximum it is instable and thus cannot be captured by X-ray

crystallography. The authors may want to use the term intermediate instead.

- In a coupled primary transporter, the change of conformations might rely on the simultaneous binding of ATP and the substrate. I thus wonder whether the authors have tried to include the substrate in their crystallization studies.
- The methods frequently refer to previous publication. Although this might be justified to shorten the length of the manuscript, key information, e.g. the detergent in which the protein was purified and crystallized should still be contained in the manuscript.
- The authors frequently describe technical details in the experimental setup and data analysis in the results section, which may be better included in the methods.
- Figure S1. Although it is reasonable to assume that Mg²⁺ is bound close to the nucleotide in the ADP-VO₄ structure, the density is not of sufficient resolution to demonstrate that.
- The lower degree of cross-linking of the A122C structure shown in Figure 3b is not apparent from the figure. The authors might want to move FigS4b to the main figures to underline their claim.

Referee #2:

Bountra et al. show a comprehensive study of the E.coli McjD ABC transporter that exports the antibacterial toxin MccJ25 encoded/produced by the expression of the other genes of the mcj operon. Through protein crystallography they present two new structures - of the ADP-VO₄ blocked enzyme similar to a previously well-determined structure of the AMPPNP complex, and of a low-resolution apoform obtained at pH 9. Interestingly and perhaps specific to McjD both forms are occluded - i.e. outward occluded ADP-VO₄ and inward-occluded apo. The structures and dynamics are further assessed by e.g. predictive Cys crosslinking, PELDOR, PEGylation, transport assays and MD simulations. Individually, most of the data are collected with proper care and controls, although the crystallographic refinement leaves some questions in particular, and the MD simulations contribute less.

The authors conclude that McjD exhibits only transient opening, which may ensure efficient release and no re-entry of the auto-toxic MccJ25.

The apo structure is new, and although determined at low resolution it appears informative to the overall conclusion of an inward-occluded form. The ADP-VO₄ structure reaffirms the observation of an outward-occluded state that further leads to the thorough qualification of an overall compact transport cycle, where open release is seemingly shortlived. Single-molecule FRET studies would be a natural next step to assess this mechanism, but would go beyond the scope of the current paper that presents an important progress and an inspiring set of observations to the field of ABC exporters.

I have a few points that should be addressed or explained:

- 1) PEGylation shows a putative intracellular opening while the crystal structure obtained at pH 9 shows a closed structure. Would it be possible to assess inward-opening dynamics or probing as a function of pH to rationalize these observations?
- 2) Crosslinking at the extracellular side prevents Hoechst33342 release (and hence liposome uptake). Will it also block binding of the Hoechst33342 to the transporter or can entrapment be shown for the crosslinked transporter?
- 3) The mPEG10K data are important, but should be discussed also with the caution that mutations may affect the structure and dynamics - what do PELDOR data or e.g. thermostability data say on the constructs used for mPEG10K mapping?
- 4) The MD simulations are of limited value and use. They are certainly not conflicting with the experimental data, which is comforting but not a strong control per se. The specific point of interest would be to pinpoint and predict the mechanism that distinguishes transient outward opening of

McjD versus stable outward opening of other exporters. Which structural elements/motifs control this function? I would consider expanding on this questions in a separate study and shorten the current MD discussion to let the experimental evidence stand out better. The manuscript would benefit from this shortening.

5) It would be interesting to expand the discussion to include the TAP peptide transport system, which shows trans-inhibition at a threshold peptide concentration (e.g. Grossmann et al. & Tampe 2014 <https://www.ncbi.nlm.nih.gov/pubmed/25377891>).

6) Table 1 is perhaps OK in the supplementary section, but given the fact that we are dealing with somewhat vague structural data figures of the electron density maps of the apo form must be included in the main text. The ADP-VO4 structure being similar to the AMPPNP structure is less important in that regard.

6b) Are the ADP-VO4 and AMPPNP crystals sufficiently isomorphous to support an Fobs(ADP-VO4) - Fobs(AMPPNP) map phased by the AMPPNP model phases? If yes, it would qualify and justify detailed discussions of small changes of the pre- and transition-state of the ATPase site.

7) Table 1 and crystal structure refinement: outer resolution bin for the ADP-VO4 form is wrong. Identical Rwork and Rfree for the ADP-VO4 is weird and indicates either mixed R-factor sets compared to the AMPPNP model structure (identical sets should be used) or improper refinement. How were B-factors handled in the low-resolution refinement of the apostructure? Hopefully not as individual B-factors - this would be an error-sink questioning the reported R-factors that are already on the edge of acceptable, even for low resolution data. R-factors should be reported with overall, single B-factor refinement or domain-by-domain single B-factor.

1st Revision - authors' response

13 July 2017

We would like to thank the referees for their positive feedback and constructive comments. We have addressed all their queries and we hope that the revised manuscript is suitable for publication. The point-by-point response to their comments are below:

Referee 1:

In the pictures it is not clear how the ADP-VO4 and apo-structures differ. A stereo figure of a superposition of CA traces would help the reader to get an impression.

We have included an additional figure that highlights the main structural differences between the vanadate and apo structures (mostly found in the NBDs) (Figure EV2).

The term outward- and inward-facing in an alternate access transporter usually refers to a distinct conformation of the transmembrane domain where a cavity leading to the substrate binding side is either oriented towards the outside or inside. Along these lines in an outward- or inward-occluded conformation, the binding site might not be accessible but the conformation should still contain features of and outward- or inward-facing states. Since the transmembrane domain is the same in all of the three known structures the authors might want to reconsider their terminology.

We would like to thank the Referee for the comment on our terminology but we believe that even though the TMD of McjD is occluded under all experimental conditions, it does share features with either inward-open or outward-open MsbA. Our group was the first to propose the terminology "outward-occluded" for McjD in the presence of AMPPNP (Choudhury *et al*, PNAS, 2014); termed outward-open due to the fact it was nucleotide bound and its TMD shared similarities with both Sav1866 and MsbA outward-open conformations with the exception of TMs1-2. Since then, this terminology is widely used on the ABC transporter field to characterize other occluded ABC exporters, PCAT1 (Lin *et al*, Nature, 2015), PglK (Perez *et al*, Nature 2015) and aaPrtd (Morgan *et al*, Structure 2017). Similarly, apo McjD shares structural features with apo exporters by having disengaged NBDs and similarities with inward-open/closed MsbA, with the exception of TMs3-5 that move towards each other to form the occluded conformation.

The authors use the term transition state for one of their structures (page 5 line 2). Since such transition state is on an energy maximum it is instable and thus cannot be captured by X-ray crystallography. The authors may want to use the term intermediate instead.

We agree with the Referee and we have changed the term high-energy transition state to high-energy intermediate state. Referee 3 made a similar comment.

In a coupled primary transporter, the change of conformations might rely on the simultaneous binding of ATP and the substrate. I thus wonder whether the authors have tried to include the substrate in their crystallization studies.

Yes, we have attempted to co-crystallise McjD with MccJ25 but we have not managed to get a co-crystal structure. We believe that it is due to a combination of the low solubility of the peptide in crystallisation solutions and the low-modest affinity of McjD at ~100 μ M (Choudhury *et al*, PNAS, 2014).

The methods frequently refer to previous publication. Although this might be justified to shorten the length of the manuscript, key information, e.g. the detergent in which the protein was purified and crystallized should still be contained in the manuscript.

We have amended the Materials and Methods section of the manuscript accordingly to include key information.

The authors frequently describe technical details in the experimental setup and data analysis in the results section, which may be better included in the methods.

We have revised the manuscript and removed technical details from the results section.

Figure S1. Although it is reasonable to assume that Mg²⁺ is bound close to the nucleotide in the ADP-VO₄ structure, the density is not of sufficient resolution to demonstrate that.

During our initial model building we calculated Fo-Fc electron density maps to verify the presence of Mg²⁺ by including ADP-VO₄ in the refinement and excluding Mg²⁺. The Fo-Fc electron density map where Mg²⁺ had been omitted from the refinement showed positive electron density. We have included a figure of this map as part of the Figure EV1.

The lower degree of cross-linking of the A122C structure shown in Figure 3b is not apparent from the figure. The authors might want to move FigS4b to the main figures to underline their claim.

Since EMBO supports Expanded View Figures, we have decided to leave the A122C histogram as part of the Expanded View Figure EV3 which will be displayed as part of Figure 3.

Referee 2:

1) PEGylation shows a putative intracellular opening while the crystal structure obtained at pH 9 shows a closed structure. Would it be possible to assess inward-opening dynamics or probing as a function of pH to rationalize these observations?

Inward-opening of the transporter is not pH dependent. We have performed the PEGylation experiment with two of the mutants, N134C and T298C, at pH9 and we do not observe reduced PEGylation, suggesting that the transporter samples an inward-opening conformation independent of pH. These data have been included in Extended View Figure EV3. This is consistent with our proposition that McjD exists mostly as an occluded conformation that samples an inward-open conformation to bind its substrate.

2) Crosslinking at the extracellular side prevents Hoechst33342 release (and hence liposome uptake). Will it also block binding of the Hoechst33342 to the transporter or can entrapment be shown for the crosslinked transporter?

It is difficult to prove if Hoechst33342 binds to cross-linked transporter but the cross-linking should not affect an inward-open conformation since the periplasmic side of the TMD aligns very well with the apo-MsbA structure.

3) The mPEG10K data are important, but should be discussed also with the caution that mutations may affect the structure and dynamics - what do PELDOR data or e.g. thermostability data say on the constructs used for mPEG10K mapping?

We have previously shown that cavity mutants do not affect the basal ATPase activity of McjD, suggesting that these mutations are not detrimental to its activity/structure (Choudhury *et al*, PNAS, 2014). Although, the ligand induced activity was affected as a result of reduced binding affinity.

4) The MD simulations are of limited value and use. They are certainly not conflicting with the experimental data, which is comforting but not a strong control per se. The specific point of interest would be to pinpoint and predict the mechanism that distinguishes transient outward opening of McjD versus stable outward opening of other exporters. Which structural elements/motifs control this function? I would consider expanding on this questions in a separate study and shorten the current MD discussion to let the experimental evidence stand out better. The manuscript would benefit from this shortening.

The MD section in the main manuscript has been shortened, as suggested by the reviewer.

5) It would be interesting to expand the discussion to include the TAP peptide transport system, which shows trans-inhibition at a threshold peptide concentration (e.g. Grossmann et al. & Tampe 2014 <https://www.ncbi.nlm.nih.gov/pubmed/25377891>).

We have included a discussion on the TAP and its trans-inhibition.

6) Table 1 is perhaps OK in the supplementary section, but given the fact that we are dealing with somewhat vague structural data figures of the electron density maps of the apo form must be included in the main text. The ADP-VO4 structure being similar to the AMPPNP structure is less important in that regard.

The electron density figures will be part of the Expanded View Figures related to Figure 1 (Figure EV1) in order to combine both answering to this remark and saving space in the Journal. We do not see the advantage of having them as main figures.

6b) Are the ADP-VO4 and AMPPNP crystals sufficiently isomorphous to support an Fobs(ADP-VO4) - Fobs(AMPPNP) map phased by the AMPPNP model phases? If yes, it would qualify and justify detailed discussions of small changes of the pre- and transition-state of the ATPase site.

The AMPPNP and ADP-VO4 crystals are not isomorphous since they belong to different space groups.

7) Table 1 and crystal structure refinement: outer resolution bin for the ADP-VO4 form is wrong. Identical Rwork and Rfree for the ADP-VO4 is weird and indicates either mixed R-factor sets compared to the AMPPNP model structure (identical sets should be used) or improper refinement. How were B-factors handled in the low-resolution refinement of the apostructure? Hopefully not as individual B-factors - this would be an error-sink questioning the reported R-factors that are already on the edge of acceptable, even for low resolution data. R-factors should be reported with overall, single B-factor refinement or domain-by-domain single B-factor.

The outer resolution bin for the ADP-VO4 has been corrected. For the ADP-VO4 data, we had to introduce new FreeR flags since the AMPPNP data are in a different spacegroup (P2₁2₁2₁) compared

to the ADP-VO4 (C2). The Rwork/Rfree have not diverged as a consequence of using a different reflection file at the end of the refinement that was processed at higher resolution and corrected for anisotropy.

The apo-structure was refined with overall B-factors. We have amended the materials and methods accordingly.

Referee 3:

1. In general, ABC exporters are classified into three types: Type I(TM287/288), Type II (ABCG5/8) and Type III(LptB2FG), The authors mainly compared the structures of McjD with MspA, but did not compare McjD with other types. Importantly, in other types of ABC exporters, for instance, the apo-ABCG5/8 and apo-LptB2FG, there are also no TMD crossovers, and NBD domains (nucleotide-bound and free forms) are not largely separate.

The focus of this manuscript was on the mechanism of type I ABC exporters whereas apo-ABCG5/8 and apo-LptB2FG belong to type II exporters. In the original manuscript we had compared TM287/288, especially at the NBDs. In the revised manuscript, we have included references to the recently published apo-ABCG5/8 and apo-LptB2FG.

2. Page 6, Paragraph 1, "We call it here high-energy transition outward-occluded." The McjD-ADP-VO4 structure is almost identical to that of the McjD-AMPPNP, why does the McjD-ADP-VO4 structure represent its high energy transition state?

Referee 1 also made a similar comment. We have changed the definition to high-energy intermediate state. The ADP-VO4 structure is a transition state mimic of a water molecule making a nucleophilic attack on the γ -phosphate of ATP, whereas the AMPPNP structure represents the ATP-bound state.

3. Page 9, Figure 2, fig. 2a and fig. 2b are in opposite position. The caption has redundant "(c)".

It has been corrected.

4. Page 10, Paragraph 2, since all L53C, A122C and S509C of the McjD can be cross-linked in the presence or absence of nucleotide, the authors should make a negative/positive control. And the control should be cysteine-less.

We have previously characterized the Cys-less McjD (Choudhury *et al*, PNAS, 2014) and we have shown that it cannot form cross-linked dimers.

5. Page 11, Paragraph 1, "In the presence of MccJ25 we observed small enhancement of A122C...", the concentration gradient of MccJ25 should be mentioned.

We have not used a concentration gradient in our cross-linking experiment. Since the amount of MccJ25 that we can produce is limited (it cannot be made synthetically due to the rotaxane topology), we chose a single concentration in order to measure an observable difference. In our original materials and methods section, we had included the concentration of MccJ25 that was used, i.e. 1 mM.

6. Page 13, Paragraph 2, "We aimed to investigate the conformational changes associated with the export of the antibacterial peptide MccJ25 from the cytoplasmic to the periplasmic leaflet of the inner membrane ", the authors should also induce a pair of R1 spin labels at the cytoplasmic side of the TMD.

We have made several mutants at the cytoplasmic side of McjD but none of them was suitable for PELDOR measurements. We have tested over 10 mutants; a few of them did not express well and others aggregated upon labelling. The cytoplasmic side of McjD contains many charged residues involved in salt-bridges that probably stabilize the overall structure, and we believe mutating them destabilizes the transporter. One of the reasons for making the predictive cysteine cross-linking

mutants was to characterize the apo occluded conformation, in the membrane, in the absence of a suitable PELDOR mutant.

Minor concerns:

1. Where there are two references cited, the "comma" sign should be superscript.

These have been corrected

2. Page 3, Introduction: Paragraph 1, line 6, "...for the surviving bacteria" should be "...for the surviving bacteria", "The microcin MccJ25(Mccj25)..." should be "The microcin J25(Mccj25)..."

The typo mistakes have been corrected.

3. Page 7, Paragraph 2, line 7, "The NBDs of the McjD-apo have separated by 7.9 Å", it'd better to be labeled in Figure 2.

We have added the distance to Fig 2.

4. Page 11, Paragraph 1, line 2, "to the AMPPNP or ADP-VO4 cross-linking experiments (Figure 3b)" should be "to the AMPPNP or ADP-VO4 cross-linking experiments (Figure 3b, Figure S3b)"

It has been added.

5. Page 15, Figure 4, b-c have no Y-axis name and scale.

We would like to thank the Referee for pointing out this; the original file that we submitted had the correct axis names and labels and there was probably an error with conversion to the pdf that we missed out. This has been corrected.

6. Page 17, Figure 5d, bands in SDS-PAGE should be labeled clearly.

We have labeled the SDS-PAGE.

2nd Editorial Decision

27 July 2017

Thank you for submitting a revised version of your manuscript. The manuscript has now been seen by all original referees, and they find that their main concerns have been addressed. There remain only a few mainly editorial issues that have to be dealt with before formal acceptance of the manuscript.

1. Please include electron density maps in the main figures, as requested by reviewer #2.
2. Please adjust references to The EMBO Journal format:
<http://emboj.embopress.org/authorguide#referencesformat>
3. Please specify panels A and B in the legend for Figure EV4
4. Figure EV5 is referred to in the text before Figure EV4, please change figure order accordingly.
5. Panels of figures 5C-D, EV1A-F, EV2A-B, EV3A-F and EV4A-B have not been referred to in the manuscript text.
6. In Figure 6C, please indicate with boxes or lines the parts of the panel that are taken from different blots.
7. Please choose five keywords (currently eight)
8. We generally encourage the publication of source data, particularly for electrophoretic gels and blots, with the aim of making primary data more accessible and transparent to the reader. We would need one file per figure (which can be a composite of source data from several panels) in jpg, gif or PDF format, uploaded as "Source data files". The gels should be labeled with the appropriate figure/panel number, and should have molecular weight markers; further annotation would clearly be useful but is not essential. These files will be published online with the article as supplementary "Source Data". Please let me know if you have any questions about this policy.

Finally, papers published in The EMBO Journal include a 'Synopsis' to further enhance discoverability. Synopses are displayed on the html version of the paper and are freely accessible to all readers. The synopsis includes a short introductory paragraph - written by the handling editor - as well as 2-5 one-sentence bullet points that summarise the paper and are provided by the authors. Please send us your suggestions for bullet points and a synopsis image. This image should provide a rapid overview of the question addressed in the study, but still needs to be kept fairly modest, since the image size cannot exceed 550x400 pixels.

Please let me know if you have any further questions regarding this or any previous points. You can use the link below to upload the revised version.

Thank you again for giving us the chance to consider your manuscript for The EMBO Journal. I am looking forward to seeing the final version.

REFEREE REPORTS

Referee #1:

I think the authors have addressed my remarks in a satisfying manner. I thus think the revised manuscript is now suitable for publication.

Referee #2:

Im quite happy with the revised version and the responses to the first review, although I think it is important to show electron density maps even though they are not pretty. This is exactly not about the aesthetics of sharing "perfect" maps (which obviously cannot be met), but rather conveying the level of information available for interpretation - and this is not much for the apo structure at low resolution with high R-factors. The reader must be led to such limitations right away through composite omit maps in the main text.

Referee #3:

The revised veriosn of this manuscript remarkably increased the readability and answered all my quesitons. I recommend to accept this paper!

2nd Revision - authors' response

03 August 2017

We have addressed all the Editorial queries and we have uploaded the revised manuscript and files.

1. Please include electron density maps in the main figures, as requested by reviewer #2.

We have moved the composite omit maps and final 2Fo-Fc electron density maps from Figure EV1 to Figure 3.

2. Please adjust references to The EMBO Journal format: <http://emboj.embopress.org/authorguide#referencesformat>

Done

The figure queries listed below have been corrected but some of the numbers have changed since we included Figure 3 from query 1.

3. Please specify panels A and B in the legend for Figure EV4

Done

4. Figure EV5 is referred to in the text before Figure EV4, please change figure order accordingly.

It has been corrected

5. Panels of figures 5C-D, EV1A-F, EV2A-B, EV3A-F and EV4A-B have not been referred to in the manuscript text.

They have been corrected

6. In Figure 6C, please indicate with boxes or lines the parts of the panel that are taken from different blots.

We have included boxes to specify the different blots.

7. Please choose five keywords (currently eight)

We have amended our keywords to 5.

8. We generally encourage the publication of source data, particularly for electrophoretic gels and blots, with the aim of making primary data more accessible and transparent to the reader. We would need one file per figure (which can be a composite of source data from several panels) in jpg, gif or PDF format, uploaded as "Source data files". The gels should be labeled with the appropriate figure/panel number, and should have molecular weight markers; further annotation would clearly be useful but is not essential. These files will be published online with the article as supplementary "Source Data". Please let me know if you have any questions about this policy.

We have included the raw blots for all the gels as Source data.

Corresponding Author Name: Konstantinos Beis

Journal Submitted to: EMBO J

Manuscript Number: EMBOJ-2017-97278